# Cue3D: Quantifying the Role of Image Cues in Single-Image 3D Generation

Xiang Li*    Zirui Wang*    Zixuan Huang    James M. Rehg

University of Illinois at Urbana-Champaign

`https://ryanxli.github.io/cue3d`

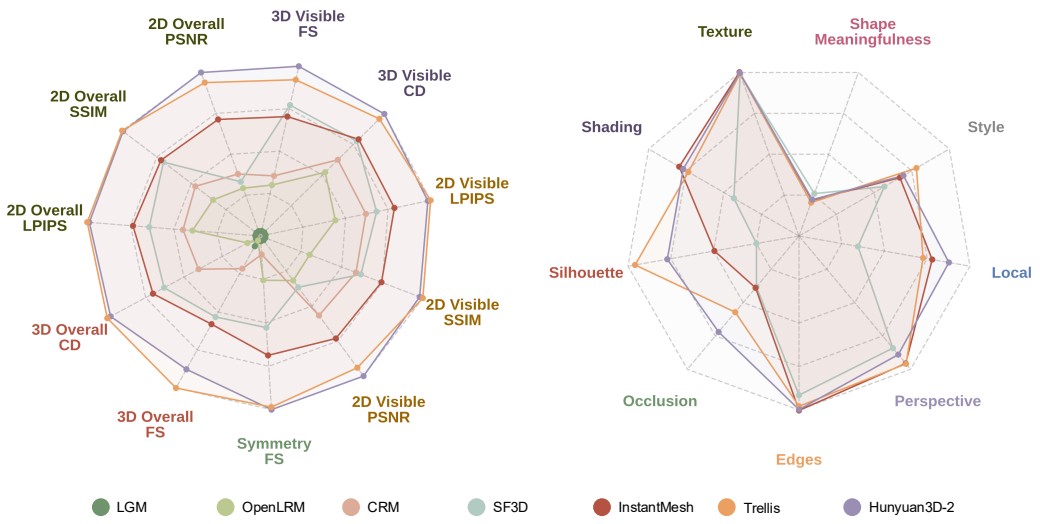

Figure 1: We present Cue3D, the first comprehensive, model-agnostic framework for quantifying the influence of individual image cues in single-image 3D generation. Left: Our unified evaluation of single-image 3D generation methods. Right: Performance robustness to the perturbation of each cue, lower values indicate higher importance. We show representative methods on Toys4K dataset for clarity; additional figures are available in the Appendix.

## Abstract

Humans and traditional computer vision methods rely on a diverse set of monocular cues to infer 3D structure from a single image, such as shading, texture, silhouette, etc. While recent deep generative models have dramatically advanced single-image 3D generation, it remains unclear which image cues these methods actually exploit. We introduce Cue3D, the first comprehensive, model-agnostic framework for quantifying the influence of individual image cues in single-image 3D generation. Our unified benchmark evaluates seven state-of-the-art methods, spanning regression-based, multi-view, and native 3D generative paradigms. By systematically perturbing cues such as shading, texture, silhouette, perspective, edges, and local continuity, we measure their impact on 3D output quality. Our analysis reveals that shape meaningfulness, not texture, dictates generalization. Geometric cues, particularly shading, are crucial for 3D generation. We further identify over-reliance on provided silhouettes and diverse sensitivities to cues such

---

*Equal contribution.

39th Conference on Neural Information Processing Systems (NeurIPS 2025).

as perspective and local continuity across model families. By dissecting these dependencies, Cue3D advances our understanding of how modern 3D networks leverage classical vision cues, and offers directions for developing more transparent, robust, and controllable single-image 3D generation models.

# 1 Introduction

Generating a 3D model from a single 2D image is a long-standing goal in computer vision, with broad applications in content creation, AR/VR, and graphics. Humans effortlessly recover 3D shape from a single view by exploiting a variety of monocular cues [3, 16, 33, 42]. Decades of research in classical computer vision studied these explicit monocular cues for shape inference, including shading patterns [21, 63], texture cues [40], silhouette contours [27], and many more. Recently, a new generation of single-image-to-3D methods has dramatically advanced the state of the art, fueled by large datasets [11] and advances in deep generative models [19]. These approaches can be grouped into three prominent categories: (i) Regression-based models that directly predict a 3D representation from the input image via a feed-forward network (*e.g.*, LRM [20], SF3D [6]), (ii) Multi-view methods that generates novel views consistent with the input image, then regress to a 3D model (*e.g.*, CRM [55], LGM [49], InstantMesh [60]), and (iii) Native 3D generative models that treat single-image-to-3D as a conditional generation problem in a learned 3D latent space (*e.g.*, Trellis [56] and Hunyuan3D-2 [64]). These approaches have enabled fast generation of textured 3D meshes from a single image, with impressive fidelity and generalization far beyond earlier methods.

Despite this rapid progress, the interpretability of single-image 3D networks remains largely under-explored. Current models are learned end-to-end on 3D supervision, and they operate as complex black boxes: we have little understanding of what information they rely on to infer 3D shape from a single image. Do these networks internally exploit the same set of visual cues as classical methods [21, 27, 40, 63], or do they rely on other information such as high-level semantics? Improving transparency in this process is important both scientifically, to connect with vision science and inform future model design, and practically, to diagnose failure modes and biases of these 3D generators.

To address this gap, we present Cue3D, the first comprehensive, model-agnostic framework for quantifying the influence of individual image cues in single-view 3D generation. We begin by establishing a unified benchmark covering seven state-of-the-art methods spanning regression-based, multi-view, and native 3D generative paradigms. We evaluate them on two standard datasets (GSO [12], Toys4K [47]). For each predicted mesh, we assess (1) both 2D appearance and 3D geometric quality for the entire shape, (2) 2D and 3D quality of the visible surface from the input viewpoint, and (3) the agreement between output and ground-truth symmetry. As summarized in Figure 1 left, native 3D generative models consistently outperform other approaches across all metrics.

We then systematically quantify the significance of each image cue. Building on meaningful perturbations [14], we disable or modify specific cues, such as silhouette shape, shading, texture semantics, perspective, and local continuity, and measure the resulting degradation in 3D output quality. Our perturbation analysis uncovers how modern single-image 3D models leverage image cues, revealing the following key insights. (1) **Meaningfulness of shape, not texture, dictates generalization.** For models to generalize, the input image must indicate a meaningful shape that does not significantly deviate from the training distribution. When we disrupt this cue by providing the models with a stochastic combination of textured primitive shapes [57], every method collapses with distinct failure modes. In contrast, the models perform surprisingly well on meaningless or random textures, with the best-performing models showing near perfect generalization. (2) **Semantics alone are not enough; Geometric cues are crucial.** Using a state-of-the-art style-transfer method [59], we convert images into artistic styles that preserve high-level semantics but often disrupts geometric cues like realistic shading and texture, as shown in Figure 2. We observe a significant drop on the performance compared to the original images, underscoring the continued importance of geometric cues. (3) **Shading is more important than texture.** To dissect the contribution of different geometric cues, we dive deeper into the image formation process. Surprisingly, even when all recognizable textures are replaced by procedural noise, natural patterns, or flat gray, for several methods, the quality of the 3D outputs remain almost unchanged, as long as the shading is kept intact. However, removing shading causes a noticeable performance decline. We further discover an interplay between shading and texture cues: intact shading alone suffices to uphold performance regardless of texture content, but when shading is removed, preserving the original texture yields better results than substituting with

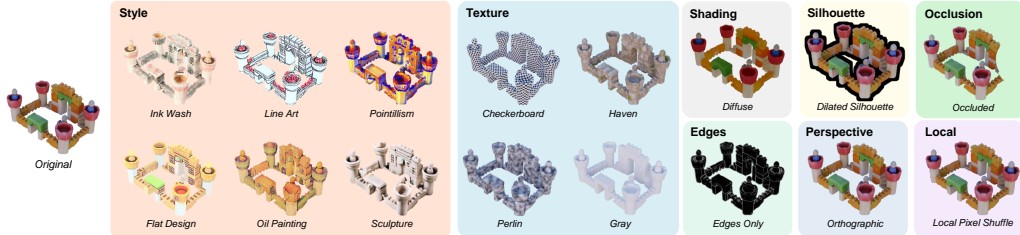

Figure 2: Overview of perturbations for analyzing individual image cues in single-image 3D generation. Starting from the original image, we systematically perturb specific visual cues. These targeted perturbations reveal the extent to which each cue influences model performance.

procedural textures or uniform color. (4) **Models are overly sensitive to silhouette and occlusion.** Dilating the object mask (without altering interior pixels) inflicts severe errors on regression-based and multi-view methods, whereas one native 3D generator remains relatively robust. In contrast, occlusion of both silhouette and image content dramatically degrades all approaches. (5) **Other cues have diverse impact.** Perturbing perspective, edge, and local-continuity signals produce measurable performance drops that vary across model categories, which we provide thorough analysis in the experiments section.

Cue3D establishes the first unified, model-agnostic framework for dissecting how modern single-image 3D generators exploit individual visual cues. Our perturbation study reveals that shape, rather than texture, meaningfulness dictates generalization. Geometric cues, especially shading, contribute significantly the 3D generation process. These models may overly rely on the provided silhouette. Meanwhile, edges, perspective, and local continuity each have distinct effects on different model families. By quantifying these dependencies across state-of-the-art approaches, Cue3D deepens our scientific understanding of image-based 3D generation, and provides potential guidance for designing more transparent, robust, and controllable single-image 3D generation methods.

## 2 Related Work

**Single-Image to 3D.** Recent advances in single-image-to-3D generation have converged on three principal paradigms. (1) Regression-based methods [6, 20, 22, 52, 54] employ neural networks to directly predict a 3D representation, such as voxels, deformed meshes, or implicit fields, from encoded image features in a single forward pass. For example, LRM [20] and its successors [6, 52] utilize transformer backbones to learn triplane representations, which are then rendered volumetrically, achieving both high fidelity and efficient inference. (2) Multi-view approaches [2, 38, 45, 49, 55, 60] follow a two-stage pipeline: first synthesizing multi-view RGB images [38], depth or coordinate maps [35, 55], normal maps [37, 39], or Gaussian splats [49], and then reconstructing 3D structure from these intermediate multi-view representations. Decoupling view synthesis from geometry enables the reuse of powerful 2D generative priors trained on billions of images [43], providing an especially strong texture prior. (3) Native 3D generative models [25, 31, 32, 53, 56, 62, 64] combine a VAE-based latent encoding of 3D data [26, 28] with a diffusion or flow model to generate high-quality and diverse 3D samples. Methods differ in their latent structures, input formats, and output representations: for instance, Hunyuan3D-2 [64] encodes point clouds to produce texture-free signed distance fields, while Trellis [56] proposes a sparse structured latent combining geometric and visual features, allowing flexible decoding into radiance fields, Gaussian splats, or meshes. Despite the iterations of approaches, it remains unclear what image cues these models rely on when producing the 3D output. In this paper, we systematically investigate how different single-image to 3D frameworks extract and transform visual signals from images cues into 3D representations.

**Image Cues.** Humans infer 3D structure from single images by integrating multiple monocular cues. Studies in developmental psychology and psychophysics show that the human visual system encodes properties like surface depth and orientation [10, 27, 46], and that internal object representations adhere to 3D constraints [44]. In contrast to humans' seamless cue integration, classical computer vision approaches explicitly leverage specific visual cues for shape inference—such as shape-from-shading [21, 23], texture gradients [24, 41], silhouettes [29, 34], contours and junctions [9], perspective effects [17], and symmetry priors [4, 51]. Modern deep models instead learn these

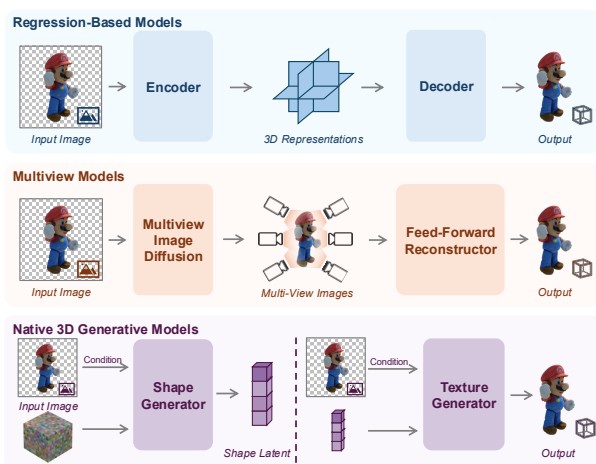
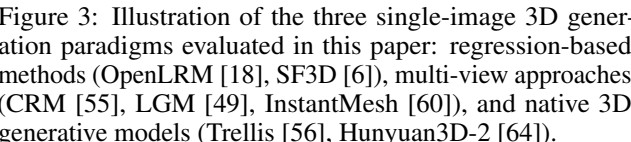

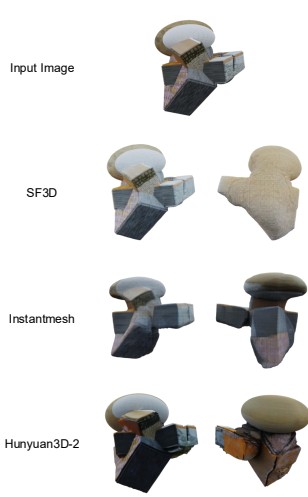

Figure 3: Illustration of the three single-image 3D generation paradigms evaluated in this paper: regression-based methods (OpenLRM [18], SF3D [6]), multi-view approaches (CRM [55], LGM [49], InstantMesh [60]), and native 3D generative models (Trellis [56], Hunyuan3D-2 [64]).

Figure 4: Qualitative comparison on the Zeroverse dataset of shapes without semantic meaning. We show one methods representative of each paradigms.

visual priors implicitly in an end-to-end manner, inspring us to explore the role of these cues in state-of-the-art models.

**Visual Cue Interpretability.** Interpreting the decision-making process of black-box models, especially with respect to the visual cues they exploit, remains an open challenge. A common strategy is input perturbation, where carefully crafted modifications are applied to input data to observe the resulting changes in model output [14, 15, 50]. For example, Geirhos et al. [15] generate images in which object shape and surface texture are semantically misaligned, revealing the relative importance of each cue in image classification models. Other approaches include latent-space probing, which trains auxiliary networks to investigate whether the internal representations of a pre-trained model fit certain downstream tasks [5, 13], and metric-based benchmarking, where models are compared across artificially curated datasets designed to emphasize specific visual attributes or cues [65]. Our work is inspired by these cue-based analysis methods but differs in key ways. Rather than solely focusing on classification or probing general features for diverse downstream tasks, we focus on presenting an in-depth analysis within the scope of single-image 3D generation. We introduce a model-agnostic framework that systematically applies controlled perturbations to distinct image cues and quantifies their effect. By evaluating a range of recent 3D architectures and employing both 2D and 3D performance metrics, our approach provides a faithful and comprehensive view of how state-of-the-art models leverage visual cues during singele-image 3D generation.

## 3 Cue3D

### 3.1 Evaluation Settings

**Methods.** We compare seven recent single-image-to-3D methods that collectively cover all three prevailing paradigms. In particular, we select OpenLRM [18] and SF3D [6] from regression-based networks, CRM [55], LGM [49] and InstantMesh [60] from multi-view reconstruction approaches, and Trellis [56] and Hunyuan3D-2 [64] from native 3D generative methods. We use the official implementation for all methods and evaluate mesh outputs in a unified way. We use 8 NVIDIA L40S GPU for all our experiments.

**Datasets.** We select two standard evaluation datasets for all methods: GSO [12], a dataset of high-quality scanned household items, and Toys4K[47], a collection of user-created 3D toy objects. We manually remove geometrically trivial objects (e.g., boxes) and balancing over-represented categories from these datasets. Our final evaluation sets contains 412 objects from the cleaned GSO dataset and 500 randomly sampled objects from the cleaned Toys4K dataset. Each object is rendered in Blender from a random camera pose (azimuth/elevation sampled within fixed limits) under a random Poly

Haven [1] HDRI lighting. More implementation details are in the appendix. To probe performance on shapes without semantic meaning, we additionally use Zeroverse [57], a procedurally generated dataset built from random assemblies of textured primitive. Zeroverse exhibits rich local geometric detail, but the shapes themselves are not meaningful, as it significantly deviate from the training distribution of single-image-to-3D methods.

**Overall Quality.** We evaluate both 2D appearance fidelity and 3D geometric quality of the 3D mesh results. We align the output mesh to the groundtruth following [6]. For appearance fidelity, we report PSNR, SSIM and LPIPS between rendered output meshes and groundtruth meshes. We render 16 views for each object with 8 uniform azimuth and 2 elevations. For geometry quality, we report the Chamfer Distance (CD) and F-scores at different thresholds to quantify the overall shape quality. The Chamfer distance between two point clouds $P_1 = \{x_i \in \mathbb{R}^3\}_{i=1}^n$ and $P_2 = \{x_j \in \mathbb{R}^3\}_{j=1}^m$. is defined as:

$$\text{chamfer}(P_1, P_2) = \frac{1}{2n} \sum_{i=1}^n |x_i - \text{NN}(x_i, P_2)| + \frac{1}{2m} \sum_{j=1}^m |x_j - \text{NN}(x_j, P_1)| \tag{1}$$

where $\text{NN}(x, P) = \arg\min_{x' \in P} \|x - x'\|$ denotes the nearest neighbor of source point $x$ in point cloud $P$.

**Visible Surface Quality.** Beyond assessing overall mesh quality, we specifically evaluate how accurately the predicted mesh aligns with the ground truth at the input image's viewpoint. We render RGB images of the output meshes from this viewpoint, obtain the corresponding depth map, and back-project the depth map into point clouds using the ground truth camera parameters. Subsequently, we employ the previously described 2D and 3D metrics on these rendered images and point clouds to quantitatively measure the quality of visible surfaces.

**Symmetry.** We further analyze the predicted object's symmetry agreement with the ground truth. Adopting the symmetry groundtruth generation procedure from [36], we apply a fixed threshold to identify planes of reflection symmetry in both predicted and ground truth meshes. For each method, we compute a binary symmetric-or-not F1 score across all predicted objects relative to their groundtruth counterparts.

### 3.2 Perturbations

We assess the importance of individual image cues through targeted perturbations. By selectively removing one cue while preserving others, significant performance degradation indicates the model's reliance on that cue. Conversely, minimal performance changes suggest the model's invariance to that cue. Additionally, preserving a single cue while removing most others can highlight its information contribution in the model's inference process. Below, we introduce the cues and their corresponding perturbations examined in this paper, illustrated in Figure 2. Additional perturbation analyses are detailed in the appendix.

**Style.** We perturb geometric cues while preserving semantic content through reference-based style transfer [59]. We select six distinct styles: ink wash, line art, pointillism, flat design, oil painting, and sculpture. We manually curate five exemplar images per style. Each object image undergoes style transfer using a randomly selected style exemplar for each of the six styles. This approach preserves core 3D structure perceptually while altering geometric cues like shading and texture, as shown in Figure 2.

**Shading & Texture.** Given their prominence as geometric cues, we jointly analyze shading and texture perturbations within the rendering pipeline. We perturb shading by rendering diffuse maps in Blender. Specifically, since the groundtruth texture in the GSO dataset has baked-in lighting, we employ an image delighting method [64] to remove baked-in lighting for the GSO dataset. Texture perturbations involve swapping original textures with alternatives such as uniform checkerboards, Perlin noise, random textures from Poly Haven [1], and uniform gray. Each texture variant is rendered both with and without lighting (diffuse).

**Silhouette and Occlusion.** Silhouette captures global shape information, and many models explicitly takes object mask as input. We investigate its influence through mask dilation and occlusion. We first dilate the silhouette (alpha mask) of each object by a fixed pixel width, leaving other cues intact. Subsequently, we simulate occlusion by placing scaled masks of randomly selected objects from the dataset onto the original mask boundaries, creating weak, medium, and strong occlusion conditions. Though occlusion partially hides image content, humans typically can still mentally reconstruct the

(a) GSO

| Method | Overall 2D | | | Overall 3D | | Symmetry | Visible Surface 2D | | | Visible Surface 3D | |
|---|---|---|---|---|---|---|---|---|---|---|---|
| | PSNR ↑ | SSIM ↑ | LPIPS ↓ | CD$_{\times 1000}$ ↓ | FS ↑ | FS ↑ | PSNR ↑ | SSIM ↑ | LPIPS ↓ | CD$_{\times 1000}$ ↓ | FS ↑ |
| LGM | 16.20 | 0.807 | 0.291 | 83.01 | 0.034 | 0.188 | 16.83 | 0.819 | 0.256 | 46.00 | 0.215 |
| OpenLRM | 17.09 | 0.820 | 0.245 | 80.89 | 0.033 | 0.391 | 17.48 | 0.824 | 0.218 | 33.00 | 0.297 |
| CRM | 17.68 | 0.833 | 0.232 | 68.07 | 0.043 | 0.285 | 18.49 | 0.845 | 0.193 | 31.10 | 0.298 |
| SF3D | 16.71 | 0.838 | 0.219 | 61.58 | 0.059 | 0.488 | 17.27 | 0.839 | 0.187 | 25.70 | 0.411 |
| InstantMesh | 19.01 | 0.849 | 0.192 | 54.54 | 0.072 | 0.715 | 19.21 | 0.853 | 0.168 | 24.00 | 0.424 |
| Hunyuan3D-2 | **19.98** | 0.862 | 0.159 | 41.82 | 0.087 | **0.894** | **20.08** | 0.863 | 0.143 | **19.10** | **0.497** |
| Trellis | 19.85 | **0.864** | **0.157** | **39.64** | **0.092** | 0.867 | 19.95 | **0.867** | **0.141** | 19.80 | 0.472 |

(b) Toys4K

| Method | Overall 2D | | | Overall 3D | | Symmetry | Visible Surface 2D | | | Visible Surface 3D | |
|---|---|---|---|---|---|---|---|---|---|---|---|
| | PSNR ↑ | SSIM ↑ | LPIPS ↓ | CD$_{\times 1000}$ ↓ | FS ↑ | FS ↑ | PSNR ↑ | SSIM ↑ | LPIPS ↓ | CD$_{\times 1000}$ ↓ | FS ↑ |
| LGM | 16.76 | 0.833 | 0.272 | 77.01 | 0.051 | 0.270 | 17.42 | 0.843 | 0.243 | 41.50 | 0.259 |
| OpenLRM | 17.85 | 0.853 | 0.221 | 74.79 | 0.047 | 0.419 | 18.51 | 0.859 | 0.192 | 28.00 | 0.351 |
| CRM | 18.21 | 0.860 | 0.214 | 61.88 | 0.064 | 0.321 | 19.45 | 0.875 | 0.170 | 25.20 | 0.370 |
| SF3D | 18.01 | 0.875 | 0.186 | 52.78 | 0.094 | 0.600 | 18.69 | 0.876 | 0.162 | 21.00 | 0.512 |
| InstantMesh | 19.59 | 0.876 | 0.173 | 49.84 | 0.098 | 0.706 | 20.06 | 0.883 | 0.149 | 20.60 | 0.489 |
| Hunyuan3D-2 | **20.79** | 0.893 | 0.138 | 38.65 | 0.126 | **0.913** | **21.08** | 0.897 | 0.124 | **14.90** | **0.590** |
| Trellis | 20.53 | **0.893** | **0.136** | **37.78** | **0.137** | 0.904 | 20.85 | **0.898** | **0.122** | 16.00 | 0.563 |

Table 1: Unified evaluation results on the GSO and Toys4K datasets. Native 3D generative models achieve the highest overall performance across metrics.

| Method | Overall 2D | | | Overall 3D | | Symmetry | Visible Surface 2D | | | Visible Surface 3D | |
|---|---|---|---|---|---|---|---|---|---|---|---|
| | PSNR ↑ | SSIM ↑ | LPIPS ↓ | CD$_{\times 1000}$ ↓ | FS ↑ | FS ↑ | PSNR ↑ | SSIM ↑ | LPIPS ↓ | CD$_{\times 1000}$ ↓ | FS ↑ |
| LGM | 16.15 | 0.754 | 0.321 | 86.19 | 0.019 | 0.000 | 16.98 | 0.767 | 0.289 | 54.50 | 0.134 |
| OpenLRM | 16.86 | 0.761 | 0.283 | 96.59 | 0.019 | 0.200 | 17.34 | 0.761 | 0.252 | 45.20 | 0.193 |
| CRM | 17.17 | 0.771 | 0.280 | 81.45 | 0.021 | 0.000 | **18.65** | **0.791** | **0.223** | 40.40 | 0.200 |
| SF3D | 15.11 | 0.767 | 0.276 | 90.34 | 0.021 | 0.267 | 15.83 | 0.768 | 0.231 | 38.60 | **0.249** |
| InstantMesh | 16.89 | 0.752 | 0.304 | 89.47 | 0.021 | 0.467 | 17.68 | 0.769 | 0.263 | 46.80 | 0.185 |
| Hunyuan3D-2 | **17.63** | 0.770 | **0.258** | **78.09** | 0.024 | **0.933** | 18.18 | 0.777 | 0.233 | **35.50** | 0.239 |
| Trellis | 17.29 | **0.773** | 0.264 | 78.14 | **0.024** | 0.467 | 17.75 | 0.781 | 0.248 | 43.20 | 0.181 |

Table 2: Evaluation results on the Zeroverse dataset of shapes without semantic meaning. Performance significantly drops compared to GSO and Toys4K, underscoring the significance of shape meaningfulness.

complete 3D shape by leveraging shape priors. These variants test the model's capability to infer complete 3D structures despite partial visibility. Additional perturbation scenarios to the silhouette are presented in the appendix.

**Edges.** Edges are analyzed due to their role in separating surfaces and indicating curvature. We first extract edge maps using the Canny algorithm from input images. Two perturbation strategies are used: one replaces all internal object cues (except silhouette) with edge maps alone, evaluating if edges can sufficiently provide information for shape inference. The other softens edges by applying Gaussian blurring only in the local neighborhood of detected edges, merging adjacent surface regions visually. Significant performance drops under these perturbations would highlight the model's reliance on precise edge information, while negligible drop would indicate invariance. Additional edge extraction methods and results are included in the appendix.

**Perspective.** Perspective cue could indicate vanishing points and spatial relationships. This cue is perturbed by switching the rendering camera to an orthographic projection. Eliminating perspective effects enables evaluation of the model's dependence on perspective cues.

**Local Continuity.** To assess sensitivity to local structural details, we perturb local continuity by splitting image foreground into grids of $n \times n$ pixels and shuffling pixels within each grid cell independently. This maintains global structure while disrupting local detail continuity. Greater performance degradation under this perturbation reflects higher sensitivity to local information.

## 4 Results

### 4.1 Unified Evaluation

We begin by conducting a unified evaluation of all seven methods on the GSO and Toys4K datasets. We present the summary of the results in Figure 1 (left), and the full evaluation details in Table 1.

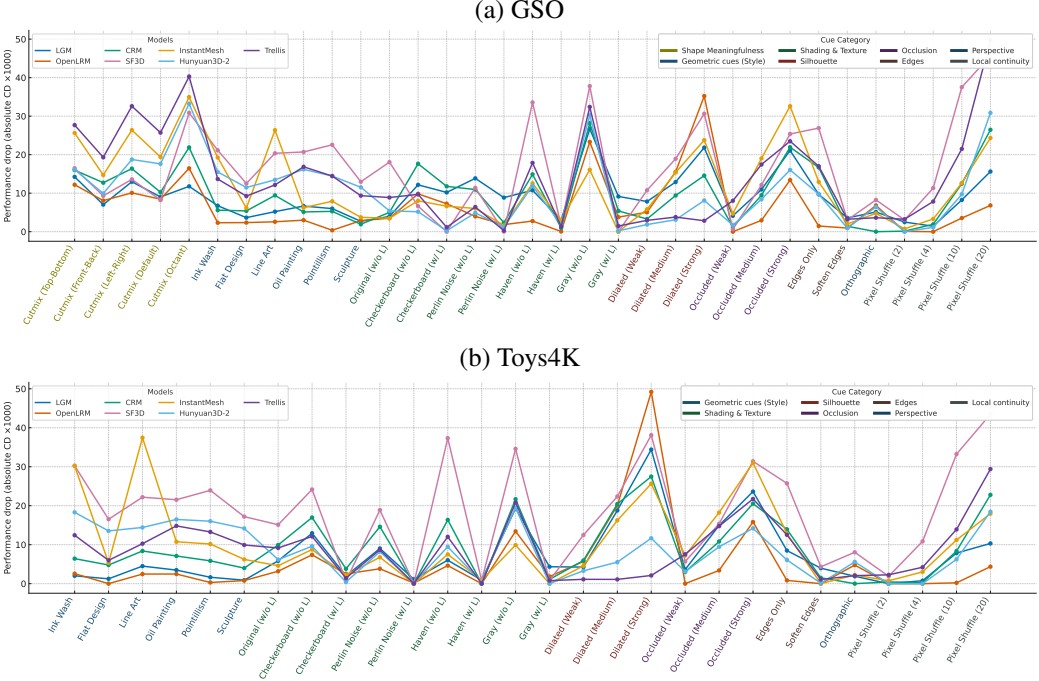

Figure 5: Quantitative analysis of image cue perturbations on single-image 3D generation. We report Chamfer Distance (CD ×1000 for clarity; lower is better) for each model under different perturbations. A larger increase in CD indicates greater performance degradation, revealing the model's reliance on the perturbed cue. A tabulated version of these results is available in Appendix.

Our results show that the native 3D generative methods, see Hunyuan3D-2 and Trellis in Table 1, clearly outperform other methods across both datasets. Generally, these two methods are closely followed by InstantMesh, the best-performing multi-view method, and SF3D, the leading regression-based method, followed by the remaining methods.

Regarding 2D appearance quality, as shown in Table 1 Overall 2D and Visible Surface 2D columns, native 3D generative methods have only marginal improvements in terms of PSNR and SSIM compared to other methods. However, they substantially outperform the alternatives in terms of LPIPS scores. This suggests that, although pixel-level statistics appear similar across methods, the native 3D generative methods more accurately capture higher-level visual information encoded by deep features.

The most substantial advantage of native 3D generative methods emerges in their 3D geometry quality, see Table 1 Overall 3D and Visible Surface 3D columns. These methods exceed the next-best method by over 10 points in overall geometry evaluation and by more than 4 points on visible surfaces under our CD×1000 metric. The visible surface quality assessments closely align with the overall geometry evaluations. Additionally, native 3D generative methods excel significantly in modeling symmetry, see Table 1 Symmetry column. They closely match the ground-truth symmetry across both datasets.

Comparing the two top methods, Hunyuan3D-2 and Trellis achieve similar 2D quality despite differences in their texture modeling approaches. Trellis demonstrates slightly better overall 3D quality, whereas Hunyuan3D-2 slightly excels in symmetry and visible surface quality. These insights provide valuable guidance for selecting the most appropriate method for practical, real-world applications.

## 4.2 Image Cues Analysis

In this section, we quantitatively analyze the role of various image cues in single-image 3D generation. An overview of our key findings is illustrated in Figure 1 (right). Detailed results are presented in Figure 5. Performances in this table are measured by Chamfer Distance (CD, scaled ×1000 for clarity), where lower values indicate better performance, thus a larger increase represents significantly

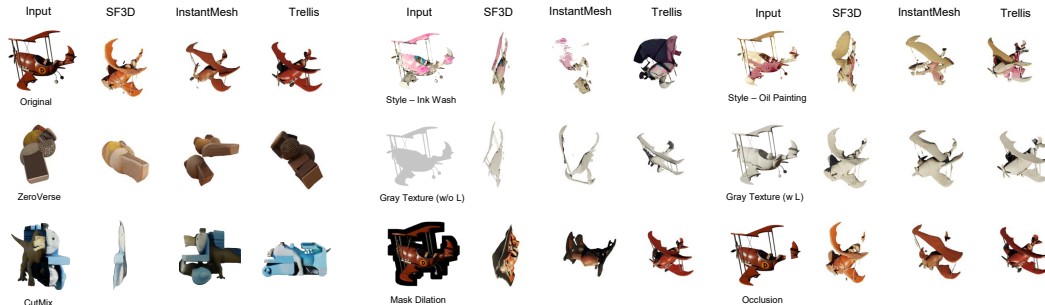

Figure 6: Qualitative example of our image cue perturbation analysis. More qualitative results are available on the project webpage and in the Appendix.

degraded performance. We show qualitative examples in Figure 6. More qualitative results are available on the project webpage and in the Appendix.

**Shape Meaningfulness.** We probe the role of shape meaningfulness in two complementary ways: (i) Zeroverse and (ii) CutMix on GSO. First, we use Zeroverse [57], a dataset comprising procedurally generated combinations of textured primitive shapes. We show qualitative results of representative methods in Figure 4, and the quantitative evaluations in Table 2. Figure 4 shows that the input image does not correspond to a meaningful shape, and has a significant gap to the training distribution. Performance notably declines across both 2D and 3D metrics compared to the more meaningful GSO and Toys4K datasets, confirming that meaningful shape in the input images are critical for the generalization of single-image 3D generation. Native 3D generative methods generally maintain the highest overall quality, while CRM best recovers visible surfaces in 2D, and SF3D and Hunyuan3D-2 perform best in visible surface 3D quality.

We further examine how the absence of shape meaningfulness influences different failure modes qualitatively in Figure 4 and quantitatively in the Appendix. Regression-based methods produce smooth, averaged back surfaces. We quantify this phenomenon by measuring the difference between each normal and the average normal in its local neighborhood, normalized against the ground truth, and we see a substantial drop of this metric on Zeroverse. Multi-view methods fail due to inconsistencies in synthesized views, as evidenced by decreased pairwise DINOv2 similarity scores, contributing to degraded 3D performance. Native 3D generative methods, lacking meaningful shape information, tend to hallucinate symmetrical completions, resulting in higher false-positive symmetry detections. See appendix for the detailed results of these experiments. These diverse failure modes underline the crucial role of meaningful shape cues, particularly for reconstructing occluded surfaces.

Meanwhile, to test shape meaningfulness with minimal domain shift, we introduce shape CutMix variants on GSO. We combine parts of different GSO meshes to perturb shape meaningfulness, while keeping appearance statistics similar. We conducted several shape CutMix experiments of varying difficulty, Given two meshes $M$ and $N$:

1) *Half-and-half*: We mix half of mesh $M$ with the other half of mesh $N$ to construct a new mesh from GSO meshes. This limits the distribution shift and preserves many local and global shape cues (e.g., surface smoothness, local symmetry), and also preserves a significant amount of shape meaningfulness to human perception. We show three variants: front-back, left-right, and top-bottom.

2) *Default CutMix*. We follow the CutMix [61] paper and randomly sample an axis-aligned 3D cube within the bounding cube of the object. We replace the part of mesh $M$ that falls into the cube with the part from mesh $N$ that falls into the same cube. When sampling the cube, we pin one of its corners at the corner of the object bounding cube to avoid significant discontinuity in the output shape. The length ratio ($\text{length}_{\text{sampled cube}}/\text{length}_{\text{bounding cube}}$) is uniformly sampled from $[0.4, 0.6]$. Most parts of the object $M$ are outside the chosen cube and remain intact. Meanwhile, the local shape cues are mostly preserved.

3) *CutMix by Octant*. We center each mesh and split it into 8 octants by the coordinate planes (XY, YZ, and XZ planes). Then we replace the part in each octant by the corresponding part from other random meshes from the same dataset. This variant still preserves the local shape cues, and it has a significantly smaller distribution gap than Zeroverse compared to our original evaluation data (GSO).

Results in Figure 5(a) show that all variants substantially hurt performance. Notably, CutMix by Octant causes a performance drop similar to Zeroverse despite a smaller domain gap. Standard CutMix, which modifies only about 1/8 of the mesh volume, still results in large drops (*e.g.*, 20 points for Hunyuan3D-2). Even minimal half-and-half perturbations, where shape meaningfulness is mostly preserved, typically cause performance drops of over 10 points, which is greater than for most other cues. This confirms that shape meaningfulness is a dominant cue for generalization.

**Geometric Cues (Style).** To explore geometric cues broadly, we apply style transfer to preserve semantics while altering geometric cues (Figure 2). Figure 5 summarizes the results on GSO and Toys4K. Performance significantly deteriorates under style perturbations. Sculpture-style images retain the most geometric information, thus yielding the smallest performance drops in general. Note that lower-performing methods show less degradation not because of their robustness, but due to metric saturation. Overall, semantics alone are insufficient; geometric cues are essential for reliable 3D inference.

**Shading and Texture.** We dissect geometric cues further by separately manipulating shading and texture, which are historically established cues for shape inference. Figure 5 presents evaluations across five texture conditions: original, checkerboard, Perlin noise, random Poly Haven texture [1], and solid gray, each tested with lighting (w/ L) and without lighting (w/o L). Surprisingly, altering texture while preserving shading minimally impacts performance for leading methods (SF3D, Hunyuan3D-2, Trellis). Multi-view approaches show slightly more sensitivity to texture changes but remain relatively robust overall. However, removing shading consistently decreases performance across methods, underscoring shading's significant role. Interestingly, there is an interplay between shading and texture cue: meaningful texture mitigates this drop due to removing lighting to some degree, especially on Toys4K. These results highlight that texture meaningfulness is not a necessary cue for generalizion. Meanwhile, shading is generally more influential than texture, with several top-performing methods exhibiting near texture invariance provided shading cues remain accurate.

**Silhouette and Occlusion.** Dilating object masks severely reduces performance despite unchanged interior pixels, indicating silhouette cues' importance. Trellis remains comparatively stable, suggesting a level of learned silhouette invariance. Occluding both silhouette and content dramatically reduces performance universally. This shows the combined importance of silhouette and interior visual cues.

**Edges.** We probe the role of edges cue in two ways, leaving only edges on silhouette and softening edges with localized gaussian filter. Edge-only input significantly degrade performance for most models except OpenLRM, suggesting edges alone may not provide sufficient shape information. Softening edges yields minor performance reductions, confirming edges are supportive but not primary cues.

**Perspective.** Switching from perspective to orthographic projection notably reduces performance, particularly for regression-based methods (OpenLRM, SF3D), indicating their reliance on perspective cues. CRM remains unaffected, since it uses orthographic images in training. Hunyuan3D-2 is more sensitive than Trellis, potentially due to its latent representation capturing perspective.

**Local Continuity.** Local cue scrambling significantly impacts regression-based SF3D, while other methods show varied but less severe sensitivity. Hunyuan3D-2 demonstrates the greatest robustness. However, all methods degrade substantially under extensive local scrambling, emphasizing the general importance of local continuity.

## 5 Discussions

**Correlation of Different Cues.** We choose our cues primarily based on their perceptual importance and interpretability to humans, rather than strict orthogonality. As noted in Section 2, our selected cues originate from psychological studies of human visual perception and have strong foundations in prior vision research, as they represent factors that humans typically find meaningful. While some cues naturally remain disentangled (*e.g.*, shading versus texture), others inherently overlap to some extent (*e.g.*, style with texture).

We assess whether cues impact objects similarly by calculating per-object performance drops in CD for each cue and then computing the Spearman rank correlation between pairs of cues. This produces a correlation matrix showing how similarly each pair of cues affects the same set of objects. We show

| | Texture | Shading | Silhouette | Occlusion | Edges | Local continuity | Style |
|---|---|---|---|---|---|---|---|
| Texture | 1.00 | 0.66 | 0.31 | 0.29 | 0.36 | 0.39 | 0.50 |
| Shading | 0.66 | 1.00 | 0.34 | 0.35 | 0.35 | 0.51 | 0.60 |
| Silhouette | 0.31 | 0.34 | 1.00 | 0.27 | 0.24 | 0.28 | 0.34 |
| Occlusion | 0.29 | 0.35 | 0.27 | 1.00 | 0.19 | 0.30 | 0.31 |
| Edges | 0.36 | 0.35 | 0.24 | 0.19 | 1.00 | 0.27 | 0.31 |
| Local cont. | 0.39 | 0.51 | 0.28 | 0.30 | 0.27 | 1.00 | 0.47 |
| Style | 0.50 | 0.60 | 0.34 | 0.31 | 0.31 | 0.47 | 1.00 |

Table 3: Analysis on the correlation of different cues. We present the Spearman rank correlations ($\rho$) between per-object performance drops in CD for each cue pair. Lower off-diagonal values indicate weaker similarity in object-wise effects; the diagonal is 1 by definition.

| Input Image | InstantMesh | SF3D | Trellis |
|---|---|---|---|
| Original | 54.5 | 61.6 | 39.6 |
| Line-art | 66.1 | 79.5 | 51.3 |
| Line-art + FLUX | 59.4 | 67.4 | 50.0 |

Table 4: Line-art-to-3D case study (lower is better). Adding explicit geometric cues to line-art narrows the gap to original images across three off-the-shelf image-to-3D models.

this result in Table 3. Importantly, this is a similarity-of-effect analysis. It does not test statistical independence nor guarantee disentanglement.

The result suggests that overall the correlation is low. Interestingly, texture and shading cues seem to affect a set of objects in similar ways, though they are inherently disentangled. These results also indicate that, while some appearance-related cues partially overlap, the cue effects are largely isolated at the level of object-wise impact.

**Practical Implications of Our Analysis.** To illustrate how our insights could inspire new research directions, we explore a line-art-to-3D problem: given a line-art image (in our case, extracted from GSO images), we aim to recover the underlying 3D shape. Pure line-art lacks surface appearance, and indeed leads to markedly worse 3D generation than original images. Inspired by our analysis, we enrich line-art with geometric cues by prompting an image diffusion model (Flux ControlNet [30]) conditioned on line-art to synthesize 3D rendering–style shading and texture. We then feed these cue-augmented images into off-the-shelf image-to-3D models (InstantMesh, SF3D, Trellis). As shown in Table 4, injecting geometric cues substantially improves performance, validating that our proposed insights could meaningfully contribute to future research in image-to-3D.

**Limitations.** While Cue3D provides a systematic and comprehensive analysis of cue importance across seven state-of-the-art methods and two widely used datasets, there remain several limitations. First, our study, though broad, is not exhaustive; evaluating a wider range of models and datasets would further strengthen our conclusions. Nevertheless, because our framework is both method and dataset-agnostic, it can be readily extended to additional settings. Second, our experiments focus on clean, object-centric datasets to minimize confounding factors, but extending the analysis to more diverse and nuanced real-world data could reveal additional insights. Third, although we primarily probe individual cues in isolation, understanding the interplay and correlation between multiple cues, beyond the initial shading-texture analysis presented here, remains an important direction for future work.

## 6   Conclusion

We introduce Cue3D, a model-agnostic framework for quantifying the influence of individual image cues in single-image 3D generation. We benchmark seven state-of-the-art methods across three major paradigms and two datasets in a unified approach. Then we apply targeted perturbations to individual cues like shading, texture, silhouette, occlusion, perspective, edges, and local continuity. We reveal that shape meaningfulness is crucial to the generalization of single-image 3D generation, while texture meaningfulness is not a necessary condition. Geometric cues are crucial, especially shading. Our analysis further shows that the models might be overly relying on silhouette cues, while perspective, edge, and local continuity cues affect reconstruction to varying degrees. We hope Cue3D and the insights presented here will deepen our understanding of how deep 3D networks leverage classical vision cues, and inspire future work on cue-aware architectures, robust training, and diagnostic perturbation tests for more transparent and controllable single-image 3D generation.

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

# 7 Additional Results on Cue3D

## 7.1 Quantitative Results

In Section 4.2 of the main paper, we discussed the results of our perturbation experiments. We present the quantitative results in Figure 5. We show the tabular version of this figure in Table 5.

**Silhouette and Edges.** In addition, we report additional perturbation results on silhouette and edges in Table 6.

For the silhouette cue, we introduce an additional perturbation called Manhattan dilation. Unlike standard uniform dilation, Manhattan dilation expands the mask boundaries to exclusively produce blocky, axis-aligned edges with each boundary line segment at least N pixels in length (where N is set to 5, 20, and 40 for our weak, medium, and strong variants, respectively). This approach further disrupts the shape information contained in the silhouette. Notably, we find that Manhattan dilation leads to a greater performance drop in regression-based and multi-view methods compared to simple dilation, whereas native 3D generative models exhibit increased robustness. This suggests that native 3D generative methods rely more heavily on extracting 3D shape cues from the overall image content, while regression-based and multi-view approaches are more overly dependent on the shape information in the silhouettes.

For the edges cue, we evaluate a range of edge extraction algorithms, including Canny [7], HED [58], Lineart [8], and PIDI [48]. While the degree of performance degradation varies across models and extraction methods, our core finding remains consistent: edge information—whether alone (combined with silhouette) is insufficient for current models to generate high-quality 3D shapes.

**Lighting and material cues.** Beyond the cues analyzed in the main text, illumination and material properties (e.g., specularity/roughness) are additional, impactful factors. We conduct a targeted study on GSO to quantify how lighting type (environment map vs. directional) and specularity level (default vs. large) affect single-image 3D reconstruction. We report Chamfer Distance (CD$\times$1000, lower is better) across six models and show deltas relative to the environment-map/default-specularity baseline. Two trends emerge: (i) directional lighting degrades performance for most models (the effect is weakest for Hunyuan3D-2), and (ii) large specularity slightly improved performance in several cases, though not always, with the exception of OpenLRM and Hunyuan3D-2. We speculate that this might be related to the different lighting setup used in each model's training.

## 7.2 Qualitative Results

We present an example of qualitative results in Figure 6 of the main paper. Comprehensive qualitative results for all perturbations are provided in Figures 9–17 at the end of the appendix. The observations from these results are consistent with our quantitative findings: native 3D generative methods generally achieve the highest quality and exhibit the greatest robustness to perturbations. Additionally, these detailed qualitative examples offer more fine-grained insights into how each perturbation affects 3D reconstruction quality for each method. Please visit our project webpage for video results.

## 7.3 Failure Modes on Zeroverse

In Table 8, 9 and Figure 8, we show different failure modes of the three model paradigms on Zeroverse, where the shapes are not meaningful. Regression-based model produces smooth back surfaces, quantified by the difference between each normal and the average normal in its local neighborhood, normalized against the groundtruth. The multi-view models cannot generate consistent

### (a) GSO

| Cue | Variant | LGM | OpenLRM | CRM | SF3D | InstantMesh | Hunyuan3D-2 | Trellis |
|---|---|---|---|---|---|---|---|---|
| Baseline | Original | 83.01 | 80.89 | 68.07 | 61.58 | 54.54 | 41.82 | 39.64 |
| Geometric Cues (Style) | Ink Wash | 89.75 (↑6.74) | 83.21 (↑2.32) | 73.66 (↑5.59) | 82.72 (↑21.14) | 73.80 (↑19.26) | 57.34 (↑15.52) | 53.31 (↑13.67) |
| | Flat Design | 86.66 (↑3.65) | 83.25 (↑2.36) | 73.46 (↑5.39) | 74.09 (↑12.51) | 60.68 (↑6.14) | 53.30 (↑11.48) | 48.87 (↑9.23) |
| | Line Art | 88.21 (↑5.20) | 83.46 (↑2.57) | 77.49 (↑9.42) | 81.94 (↑20.36) | 80.90 (↑26.36) | 55.27 (↑13.45) | 51.79 (↑12.15) |
| | Oil Painting | 89.66 (↑6.65) | 83.88 (↑2.99) | 73.19 (↑5.12) | 82.28 (↑20.70) | 60.77 (↑6.23) | 58.01 (↑16.19) | 56.49 (↑16.85) |
| | Pointillism | 89.00 (↑5.99) | 81.26 (↑0.37) | 73.35 (↑5.28) | 84.14 (↑22.56) | 62.44 (↑7.90) | 56.26 (↑14.44) | 54.08 (↑14.44) |
| | Sculpture | 85.70 (↑2.69) | 83.76 (↑2.87) | 69.99 (↑1.92) | 74.54 (↑12.96) | 58.27 (↑3.73) | 53.35 (↑11.53) | 48.99 (↑9.35) |
| Shading & Texture | Original (w/o L) | 86.97 (↑3.96) | 84.38 (↑3.49) | 73.08 (↑5.01) | 79.64 (↑18.06) | 58.02 (↑3.48) | 47.22 (↑5.40) | 48.51 (↑8.87) |
| | Checkerboard (w/o L) | 95.17 (↑12.16) | 90.71 (↑9.82) | 85.71 (↑17.64) | 68.20 (↑6.62) | 62.55 (↑8.01) | 47.00 (↑5.18) | 49.33 (↑9.69) |
| | Checkerboard (w/ L) | 93.21 (↑10.20) | 88.15 (↑7.26) | 79.84 (↑11.77) | 61.19 (↓0.39) | 61.16 (↑6.62) | 42.06 (↑0.24) | 40.78 (↑1.14) |
| | Perlin Noise (w/o L) | 96.86 (↑13.85) | 84.86 (↑3.97) | 79.01 (↑10.94) | 72.97 (↑11.39) | 60.57 (↑6.03) | 46.70 (↑4.88) | 46.06 (↑6.42) |
| | Perlin Noise (w/ L) | 91.86 (↑8.85) | 82.72 (↑1.83) | 70.40 (↑2.33) | 61.62 (↑0.04) | 56.16 (↑1.62) | 43.06 (↑1.24) | 39.91 (↑0.27) |
| | Haven (w/o L) | 93.76 (↑10.75) | 83.65 (↑2.76) | 82.96 (↑14.89) | 95.15 (↑33.57) | 67.09 (↑12.55) | 53.73 (↑11.91) | 57.51 (↑17.87) |
| | Haven (w/ L) | 86.01 (↑3.00) | 80.94 (↑0.05) | 69.02 (↑0.95) | 63.11 (↑1.53) | 57.57 (↑3.03) | 43.62 (↑1.80) | 40.97 (↑1.33) |
| | Gray (w/o L) | 109.70 (↑26.69) | 104.22 (↑23.33) | 96.26 (↑28.19) | 99.38 (↑37.80) | 70.63 (↑16.09) | 72.12 (↑30.30) | 72.04 (↑32.40) |
| | Gray (w/ L) | 92.16 (↑9.15) | 84.67 (↑3.78) | 73.46 (↑5.39) | 61.13 (↓0.45) | 54.52 (↓0.02) | 42.05 (↑0.23) | 41.11 (↑1.47) |
| Silhouette | Dilated (Weak) | 90.84 (↑7.83) | 85.94 (↑5.05) | 71.36 (↑3.29) | 72.34 (↑10.76) | 60.35 (↑5.81) | 43.67 (↑1.85) | 42.58 (↑2.94) |
| | Dilated (Medium) | 95.90 (↑12.89) | 96.42 (↑15.53) | 77.47 (↑9.40) | 80.51 (↑18.93) | 69.92 (↑15.38) | 44.92 (↑3.10) | 43.43 (↑3.79) |
| | Dilated (Strong) | 104.84 (↑21.83) | 116.11 (↑35.22) | 82.63 (↑14.56) | 92.20 (↑30.62) | 78.31 (↑23.77) | 49.92 (↑8.10) | 42.48 (↑2.84) |
| Occlusion | Occluded (Weak) | 87.13 (↑4.12) | 79.54 (↓1.35) | 69.25 (↑1.18) | 62.23 (↑0.65) | 59.22 (↑4.68) | 43.51 (↑1.69) | 47.67 (↑8.03) |
| | Occluded (Medium) | 93.97 (↑10.96) | 83.83 (↑2.94) | 77.53 (↑9.46) | 73.70 (↑12.12) | 73.57 (↑19.03) | 50.23 (↑8.41) | 57.09 (↑17.45) |
| | Occluded (Strong) | 104.10 (↑21.09) | 94.30 (↑13.41) | 90.01 (↑21.94) | 86.95 (↑25.37) | 87.13 (↑32.59) | 57.86 (↑16.04) | 63.13 (↑23.49) |
| Edges | Edges Only | 92.70 (↑9.69) | 82.36 (↑1.47) | 84.74 (↑16.67) | 88.48 (↑26.90) | 67.47 (↑12.93) | 51.56 (↑9.74) | 56.64 (↑17.00) |
| | Soften Edges | 86.58 (↑3.57) | 81.84 (↑0.95) | 69.47 (↑1.40) | 64.63 (↑3.05) | 56.54 (↑2.00) | 42.76 (↑0.94) | 42.87 (↑3.23) |
| Perspective | Orthographic | 88.12 (↑5.11) | 87.70 (↑6.81) | 66.25 (↓1.82) | 69.83 (↑8.25) | 59.30 (↑4.76) | 48.24 (↑6.42) | 43.26 (↑3.62) |
| Local continuity | Pixel Shuffle (2) | 85.50 (↑2.49) | 80.97 (↑0.08) | 68.21 (↑0.14) | 64.62 (↑3.04) | 55.27 (↑0.73) | 41.12 (↓0.70) | 42.87 (↑3.23) |
| | Pixel Shuffle (4) | 84.43 (↑1.42) | 80.34 (↓0.55) | 69.98 (↑1.91) | 72.90 (↑11.32) | 57.85 (↑3.31) | 42.94 (↑1.12) | 47.47 (↑7.83) |
| | Pixel Shuffle (10) | 91.25 (↑8.24) | 84.42 (↑3.53) | 80.53 (↑12.46) | 99.10 (↑37.52) | 67.17 (↑12.63) | 51.54 (↑9.72) | 61.13 (↑21.49) |
| | Pixel Shuffle (20) | 98.63 (↑15.62) | 87.71 (↑6.82) | 94.53 (↑26.46) | 107.32 (↑45.74) | 78.88 (↑24.34) | 72.68 (↑30.86) | 89.44 (↑49.80) |

### (b) Toys4K

| Cue | Variant | LGM | OpenLRM | CRM | SF3D | InstantMesh | Hunyuan3D-2 | Trellis |
|---|---|---|---|---|---|---|---|---|
| Baseline | Original | 77.01 | 74.79 | 61.88 | 52.78 | 49.84 | 38.65 | 37.78 |
| Geometric Cues (Style) | Ink Wash | 79.02 (↑2.01) | 77.36 (↑2.57) | 68.30 (↑6.42) | 83.03 (↑30.25) | 80.04 (↑30.20) | 56.95 (↑18.30) | 50.20 (↑12.42) |
| | Flat Design | 78.25 (↑1.24) | 74.64 (↓0.15) | 66.66 (↑4.78) | 69.30 (↑16.52) | 69.30 (↑19.46) | 55.19 (↑13.53) | 43.76 (↑5.98) |
| | Line Art | 81.51 (↑4.50) | 77.25 (↑2.46) | 70.26 (↑8.38) | 74.96 (↑22.18) | 87.29 (↑37.45) | 53.07 (↑14.42) | 48.04 (↑10.26) |
| | Oil Painting | 80.49 (↑3.48) | 77.26 (↑2.47) | 68.98 (↑7.10) | 74.31 (↑21.53) | 60.59 (↑10.75) | 55.13 (↑16.48) | 52.60 (↑14.82) |
| | Pointillism | 78.65 (↑1.64) | 75.09 (↑0.30) | 67.73 (↑5.85) | 76.72 (↑23.94) | 60.01 (↑10.17) | 54.67 (↑16.02) | 51.05 (↑13.27) |
| | Sculpture | 77.90 (↑0.89) | 75.58 (↑0.79) | 65.87 (↑3.99) | 69.98 (↑17.20) | 56.06 (↑6.22) | 52.82 (↑14.17) | 47.70 (↑9.92) |
| Shading & Texture | Original (w/o L) | 82.86 (↑5.85) | 77.98 (↑3.19) | 71.80 (↑9.92) | 67.89 (↑15.11) | 54.44 (↑4.60) | 44.81 (↑6.16) | 46.94 (↑9.16) |
| | Checkerboard (w/o L) | 89.95 (↑12.94) | 82.18 (↑7.39) | 78.85 (↑16.97) | 76.94 (↑24.16) | 58.60 (↑8.76) | 48.26 (↑9.61) | 49.89 (↑12.11) |
| | Checkerboard (w/ L) | 78.98 (↑1.97) | 77.30 (↑2.51) | 65.68 (↑3.80) | 53.46 (↑0.68) | 52.03 (↑2.19) | 39.06 (↑0.41) | 39.19 (↑1.41) |
| | Perlin Noise (w/o L) | 86.04 (↑9.03) | 78.61 (↑3.82) | 76.49 (↑14.61) | 71.67 (↑18.89) | 56.61 (↑6.77) | 46.85 (↑8.20) | 46.48 (↑8.70) |
| | Perlin Noise (w/ L) | 78.14 (↑1.13) | 74.94 (↑0.15) | 61.19 (↓0.69) | 52.43 (↓0.35) | 49.16 (↓0.68) | 38.79 (↑0.14) | 37.11 (↓0.67) |
| | Haven (w/o L) | 83.03 (↑6.02) | 79.42 (↑4.63) | 78.24 (↑16.36) | 90.11 (↑37.33) | 57.35 (↑7.51) | 48.10 (↑9.45) | 49.80 (↑12.02) |
| | Haven (w/ L) | 77.90 (↑0.89) | 74.67 (↓0.12) | 61.76 (↓0.12) | 53.56 (↑0.78) | 50.77 (↑0.93) | 39.09 (↑0.44) | 37.81 (↑0.03) |
| | Gray (w/o L) | 96.32 (↑19.31) | 88.22 (↑13.43) | 83.56 (↑21.68) | 87.37 (↑34.59) | 59.73 (↑9.89) | 58.12 (↑19.47) | 58.62 (↑20.84) |
| | Gray (w/ L) | 81.35 (↑4.34) | 76.62 (↑1.83) | 62.96 (↑1.08) | 53.55 (↑0.77) | 47.54 (↓2.30) | 38.28 (↓0.37) | 38.56 (↑0.78) |
| Silhouette | Dilated (Weak) | 81.26 (↑4.25) | 80.48 (↑5.69) | 67.82 (↑5.94) | 65.25 (↑12.47) | 54.41 (↑4.57) | 41.94 (↑3.29) | 38.89 (↑1.11) |
| | Dilated (Medium) | 95.79 (↑18.78) | 94.80 (↑20.01) | 82.32 (↑20.44) | 75.15 (↑22.37) | 66.09 (↑16.25) | 44.17 (↑5.52) | 38.87 (↑1.09) |
| | Dilated (Strong) | 111.42 (↑34.41) | 123.99 (↑49.20) | 89.36 (↑27.48) | 90.88 (↑38.10) | 75.48 (↑25.64) | 50.30 (↑11.65) | 39.88 (↑2.10) |
| Occlusion | Occluded (Weak) | 80.63 (↑3.62) | 73.98 (↓0.81) | 64.87 (↑2.99) | 58.46 (↑5.68) | 57.06 (↑7.22) | 41.74 (↑3.09) | 45.31 (↑7.53) |
| | Occluded (Medium) | 91.94 (↑14.93) | 78.18 (↑3.39) | 72.69 (↑10.81) | 68.14 (↑15.36) | 68.08 (↑18.24) | 48.09 (↑9.44) | 52.54 (↑14.76) |
| | Occluded (Strong) | 100.64 (↑23.63) | 90.62 (↑15.83) | 82.41 (↑20.53) | 84.17 (↑31.39) | 80.83 (↑30.99) | 52.84 (↑14.19) | 59.51 (↑21.73) |
| Edges | Edges Only | 85.50 (↑8.49) | 75.63 (↑0.84) | 75.83 (↑13.95) | 78.51 (↑25.73) | 62.68 (↑12.84) | 44.74 (↑6.09) | 50.31 (↑12.53) |
| | Soften Edges | 80.98 (↑3.97) | 74.33 (↓0.46) | 63.37 (↑1.49) | 57.10 (↑4.32) | 50.77 (↑0.93) | 38.77 (↑0.12) | 38.86 (↑1.08) |
| Perspective | Orthographic | 78.91 (↑1.90) | 79.56 (↑4.77) | 60.31 (↓1.57) | 60.83 (↑8.05) | 52.06 (↑2.22) | 44.24 (↑5.59) | 39.84 (↑2.06) |
| Local continuity | Pixel Shuffle (2) | 76.69 (↓0.32) | 74.45 (↓0.34) | 61.44 (↓0.44) | 54.67 (↑1.89) | 50.58 (↑0.74) | 37.59 (↓1.06) | 40.04 (↑2.26) |
| | Pixel Shuffle (4) | 77.77 (↑0.76) | 73.28 (↓1.51) | 61.60 (↓0.28) | 63.64 (↑10.86) | 52.86 (↑3.02) | 37.96 (↓0.69) | 41.99 (↑4.21) |
| | Pixel Shuffle (10) | 84.86 (↑7.85) | 74.58 (↓0.21) | 70.24 (↑8.36) | 86.01 (↑33.23) | 61.05 (↑11.21) | 44.90 (↑6.25) | 51.69 (↑13.91) |
| | Pixel Shuffle (20) | 87.33 (↑10.32) | 79.15 (↑4.36) | 84.66 (↑22.78) | 96.24 (↑43.46) | 67.80 (↑17.96) | 57.11 (↑18.46) | 67.18 (↑29.40) |

Table 5: Quantitative analysis of image cue perturbations on single-image 3D generation. We report Chamfer Distance (CD ×1000 for clarity; lower is better) for each model under different perturbations. A larger increase in CD indicates greater performance degradation, revealing the model's reliance on the perturbed cue.

views, shown by the dropping DINOv2 similarity across views. Native 3D generative models hallucinate non-existent symmetries, shown by a large number of false positives on Zeroverse.

## 7.4 Variance of Different Seeds in Viewpoint Sampling

We observe minor performance differences when evaluating the models using the same input image using different random seeds. However, when we vary the random seeds to sample different viewpoints of the same object, thus generating different rendered input images, some performance variation emerges, as reported in Table 10. Importantly, these variations do not affect the core findings or conclusions of our study. A more comprehensive investigation into viewpoint sensitivity remains an interesting direction for future work.

#### (a) GSO

| Cue | Variant | LGM | OpenLRM | CRM | SF3D | InstantMesh | Hunyuan3D-2 | Trellis |
|---|---|---|---|---|---|---|---|---|
| Baseline | Original | 83.01 | 80.89 | 68.07 | 61.58 | 54.54 | 41.82 | 39.64 |
| Silhouette | Manhattan Dilated (Weak) | 89.80 (↑6.79) | 82.69 (↑1.80) | 69.97 (↑1.90) | 73.01 (↑11.43) | 57.58 (↑3.04) | 40.94 (↓0.88) | 41.91 (↑2.27) |
| | Manhattan Dilated (Mid) | 103.99 (↑20.98) | 96.19 (↑15.30) | 79.50 (↑11.43) | 89.98 (↑28.40) | 70.07 (↑15.53) | 42.70 (↑0.88) | 41.16 (↑1.52) |
| | Manhattan Dilated (Strong) | 109.75 (↑26.74) | 107.11 (↑26.22) | 84.28 (↑16.21) | 97.01 (↑35.43) | 82.75 (↑28.21) | 45.28 (↑3.46) | 41.11 (↑1.47) |
| Edges | Canny Edges | 92.70 (↑9.69) | 82.36 (↑1.47) | 84.74 (↑16.67) | 88.48 (↑26.90) | 67.47 (↑12.93) | 51.56 (↑9.74) | 56.64 (↑17.00) |
| | HED Edges | 95.63 (↑12.62) | 84.31 (↑3.42) | 83.27 (↑15.20) | 96.18 (↑34.60) | 70.25 (↑15.71) | 48.85 (↑7.03) | 55.53 (↑15.89) |
| | Lineart Edges | 90.23 (↑7.22) | 85.61 (↑4.72) | 79.39 (↑11.32) | 79.53 (↑17.95) | 66.11 (↑11.57) | 50.37 (↑8.55) | 51.30 (↑11.66) |
| | PIDI Edges | 94.25 (↑11.24) | 84.45 (↑3.56) | 84.13 (↑16.06) | 102.63 (↑41.05) | 73.17 (↑18.63) | 49.06 (↑7.24) | 69.06 (↑29.42) |

#### (b) Toys4K

| Cue | Variant | LGM | OpenLRM | CRM | SF3D | InstantMesh | Hunyuan3D-2 | Trellis |
|---|---|---|---|---|---|---|---|---|
| Baseline | Original | 77.01 | 74.79 | 61.88 | 52.78 | 49.84 | 38.65 | 37.78 |
| Silhouette | Manhattan Dilated (Weak) | 85.71 (↑8.70) | 77.97 (↑3.18) | 65.13 (↑3.25) | 64.46 (↑11.68) | 53.66 (↑3.82) | 39.13 (↑0.48) | 39.59 (↑1.81) |
| | Manhattan Dilated (Mid) | 104.77 (↑27.76) | 93.12 (↑18.33) | 76.56 (↑14.68) | 83.10 (↑30.32) | 68.84 (↑19.00) | 38.61 (↓0.04) | 38.40 (↑0.62) |
| | Manhattan Dilated (Strong) | 117.13 (↑40.12) | 108.18 (↑33.39) | 90.44 (↑28.56) | 91.02 (↑38.24) | 82.37 (↑32.53) | 42.09 (↑3.44) | 38.75 (↑0.97) |
| Edges | Canny Edges | 85.50 (↑8.49) | 75.63 (↑0.84) | 75.83 (↑13.95) | 78.51 (↑25.73) | 62.68 (↑12.84) | 44.74 (↑6.09) | 50.31 (↑12.53) |
| | HED Edges | 88.10 (↑11.09) | 75.42 (↑0.63) | 76.21 (↑14.33) | 88.75 (↑35.97) | 66.70 (↑16.86) | 43.63 (↑4.98) | 51.29 (↑13.51) |
| | Lineart Edges | 82.90 (↑5.89) | 76.06 (↑1.27) | 72.83 (↑10.95) | 72.33 (↑19.55) | 62.76 (↑12.92) | 48.79 (↑10.14) | 46.61 (↑8.83) |
| | PIDI Edges | 89.81 (↑12.80) | 76.35 (↑1.56) | 76.10 (↑14.22) | 93.51 (↑40.73) | 69.67 (↑19.83) | 43.57 (↑4.92) | 64.70 (↑26.92) |

Table 6: Additional Quantitative analysis of image cue perturbations on single-image 3D generation. We report Chamfer Distance (CD ×1000; lower is better) for each model under different perturbations. A larger increase in CD indicates greater performance degradation, revealing the model's reliance on the perturbed cue.

| Lighting & Material Variants | OpenLRM | CRM | SF3D | InstantMesh | Hunyuan3D-2 | Trellis |
|---|---|---|---|---|---|---|
| Env map, default specularity (orig.) | 80.89 | 68.07 | 61.58 | 54.54 | 41.82 | 39.64 |
| Env map, large specularity | 81.70 (+0.81) | 68.40 (+0.33) | 61.84 (+0.26) | 54.43 (−0.11) | 42.32 (+0.50) | 38.94 (−0.70) |
| Directional, default specularity | 84.19 (+3.30) | 72.00 (+3.93) | 67.26 (+5.68) | 60.60 (+6.06) | 42.77 (+0.95) | 43.97 (+4.33) |
| Directional, large specularity | 84.47 (+3.58) | 70.68 (+2.61) | 65.99 (+4.41) | 57.94 (+3.40) | 42.76 (+0.94) | 42.54 (+2.90) |

Table 7: GSO ablation on lighting and specularity (CD×1000, lower is better). Values in parentheses are deltas vs. the environment-map/default-specularity baseline. Directional lighting generally hurts, while increased specularity can mitigate or improve performance depending on model and lighting.

## 8 Implementation Details

### 8.1 Evaluation

We first leverage ambient occlusion to remove the internal surface of the output meshes. For each predicted mesh, we load and recenter both the prediction and its corresponding ground-truth mesh, normalizing each to a unit bounding sphere. Following [6], we uniformly search a dense grid of rotations (24 azimuth × 24 elevation × 12 roll samples), applies each to the predicted cloud, and perform a brute-force search over these rotations to identify the best coarse alignment. Finally, we refine this alignment with Iterative Closest Point (ICP). After the alignment, we compute Chamfer Distance (CD) and F-score at CD thresholds to evaluate the predicted meshes.

### 8.2 Perturbations

Our input images are 512 × 512 pixels in resolution. For silhouette dilation, we apply a dilation kernel of 10 pixels for the weak variant, 30 pixels for the medium variant, and 60 pixels for the strong variant. For occlusion, we randomly position an occluder mask along the edge of the object and scale it by a factor of 0.1, 0.4, or 0.8 for the weak, medium, and strong variants, respectively. For pixel shuffle, we randomly shuffle all pixels within each non-overlapping N × N grid inside the object mask, with N set to 2, 4, 10, or 20 to represent different perturbation strength.

### 8.3 Teaser Figure

In Figure 1 of the main paper, we provide an overview of our key findings, including performance comparisons and robustness to perturbations, on the Toys4K dataset. Here, we present analogous figures for the GSO dataset, where the results closely mirror those observed on Toys4K.

The radar plot on the right illustrates robustness to different image cues. Each axis shows the increase in Chamfer Distance (CD) of using a perturbed image relative to using the original image, with values normalized from 0 to 1 according to the largest drop across all models and cues. For the texture axis, we report the average performance drop over all texture-swap perturbations with

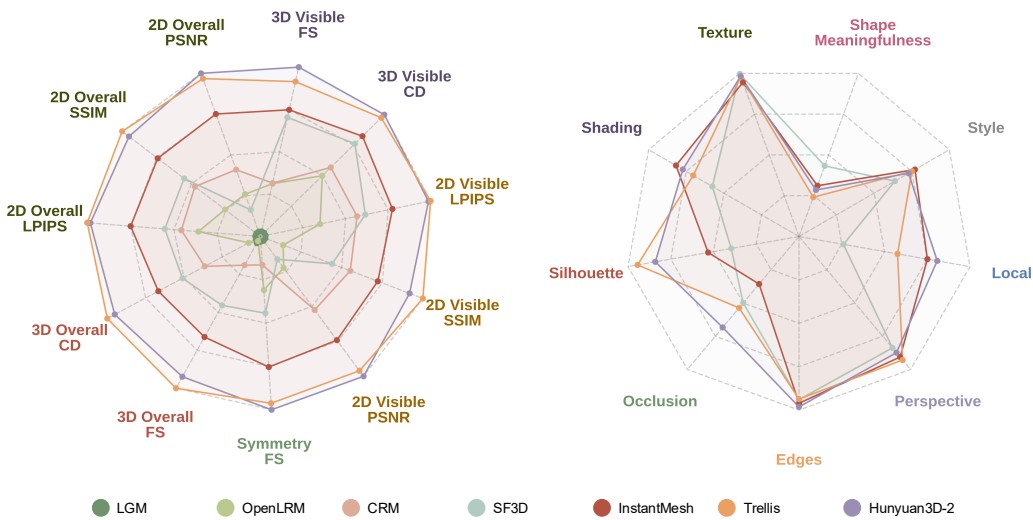

Figure 7: Teaser figure result on the GSO dataset. We observe the same trend as in Toys4K. Left: Our unified evaluation of single-image 3D generation methods. Right: Performance robustness to the perturbation of each cue, lower values indicate higher importance.

| Method | GSO | Toys4K | Zeroverse |
|--------|-----|--------|-----------|
| OpenLRM | 1.5819 | 1.3361 | 0.7892 |
| SF3D | 1.0198 | 0.9155 | 0.3997 |

Table 8: Failure mode of regression-based models on meaningless shapes: back view becomes smooth, measured by a normal roughness index, lower means smoother.

| Method | GSO | Toys4K | Zeroverse |
|--------|-----|--------|-----------|
| CRM | 0.5214 | 0.5583 | 0.4786 |
| InstantMesh | 0.6628 | 0.7462 | 0.5348 |

Table 9: Failure mode of multi-view models on meaningless shapes: view inconsistency, measured by DINOv2 similarity.

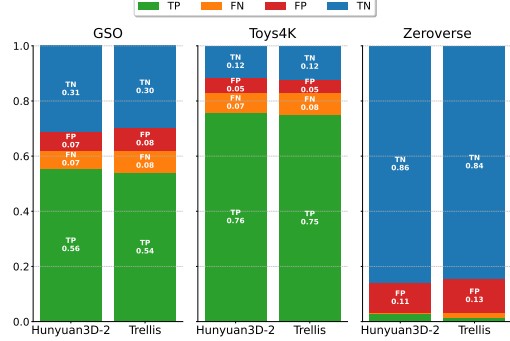

Figure 8: Failure mode of native 3D generative models on meaningless shapes: hallucination of symmetry, shown by the increase of false positives on Zeroverse.

shading intact; for shading, we average over all shading-related perturbations without lighting. For silhouette and occlusion, we use the strong perturbation variant. For edges, we use the softened edges perturbation; for local continuity, we use the strength of 10; for style, we average across all styles; and for shape meaningfulness, we report the drop compared to Zeroverse performance. We display only the best-performing methods because, for weaker models, the original CD scores are poor, our ICP alignment can lead to CD scores saturating, making CD score drop no longer able to faithfully represent performance degradation, or no longer comparable with other better-performing models.

# 9 Broader Impact

On the positive side, Cue3D addresses a fundamental gap in the interpretability and robustness of single-image 3D generation models. By systematically dissecting which visual cues, such as shading, texture, silhouette, and perspective, that are actually used by state-of-the-art 3D generative networks, this work lays the foundation for developing more transparent, robust, and controllable AI systems. Such advances have broad societal benefits. For example, in content creation and digital entertainment, better understanding of 3D model generation can drive higher quality, more reliable virtual assets for animation, games, AR/VR, and industrial design. In scientific and educational settings, robust

| Method | SF3D | InstantMesh | Trellis |
|---|---|---|---|
| Seed 1 | 61.58 | 54.54 | 39.64 |
| Seed 2 | 58.00 | 56.45 | 39.70 |
| Seed 3 | 58.20 | 53.88 | 40.52 |
| Average | 59.26 | 54.96 | 39.95 |
| Standard Deviation | 1.64 | 1.09 | 0.40 |

Table 10: Performance variation when using different seeds to sample inference images viewpoint, measured by Chamfer Distance (CD ×1000; lower is better).

3D inference from images can facilitate improved visualization and analysis of objects and scenes, democratizing access to powerful graphics tools. More broadly, the push for interpretability aligns with growing public and regulatory demand for transparency and trustworthiness in AI, reducing the risk of unpredictable failures and biases in downstream applications. The open, extensible benchmarking methodology promoted by Cue3D could become a community standard, encouraging more responsible, reproducible, and diagnosable progress in AI-generated 3D content.

However, there are also potential negative societal impacts. As single-image 3D generation methods become more powerful and transparent, the same advances that benefit creative and scientific communities could be leveraged for malicious purposes. Improved 3D reconstruction from ordinary images can facilitate unauthorized cloning of real-world objects, cultural artifacts, or even biometric data (such as faces or bodies), raising concerns about privacy, copyright infringement, and the propagation of deepfakes. The increased robustness of these models to stylization or occlusion, as discussed in the paper, could make it easier to reconstruct 3D models even from intentionally obfuscated or partially hidden images, weakening existing privacy protections. Moreover, because the findings highlight the over-reliance on certain cues (e.g., silhouette), there is a risk that future systems, if not carefully designed, could propagate existing biases or vulnerabilities into real-world deployments, for example, failing more frequently on out-of-distribution or less-represented shapes, which may disproportionately impact marginalized communities or less commonly encountered objects. The computational demands of benchmarking and training these systems also carry environmental costs, which, while not unique to this work, are worth considering given the scale of modern AI experiments.

In summary, while Cue3D represents an important step toward more interpretable, robust, and community-driven single-image 3D generation, care must be taken to address privacy, bias, and misuse risks, ensuring that these technological advances are ultimately used for societal benefit rather than harm.

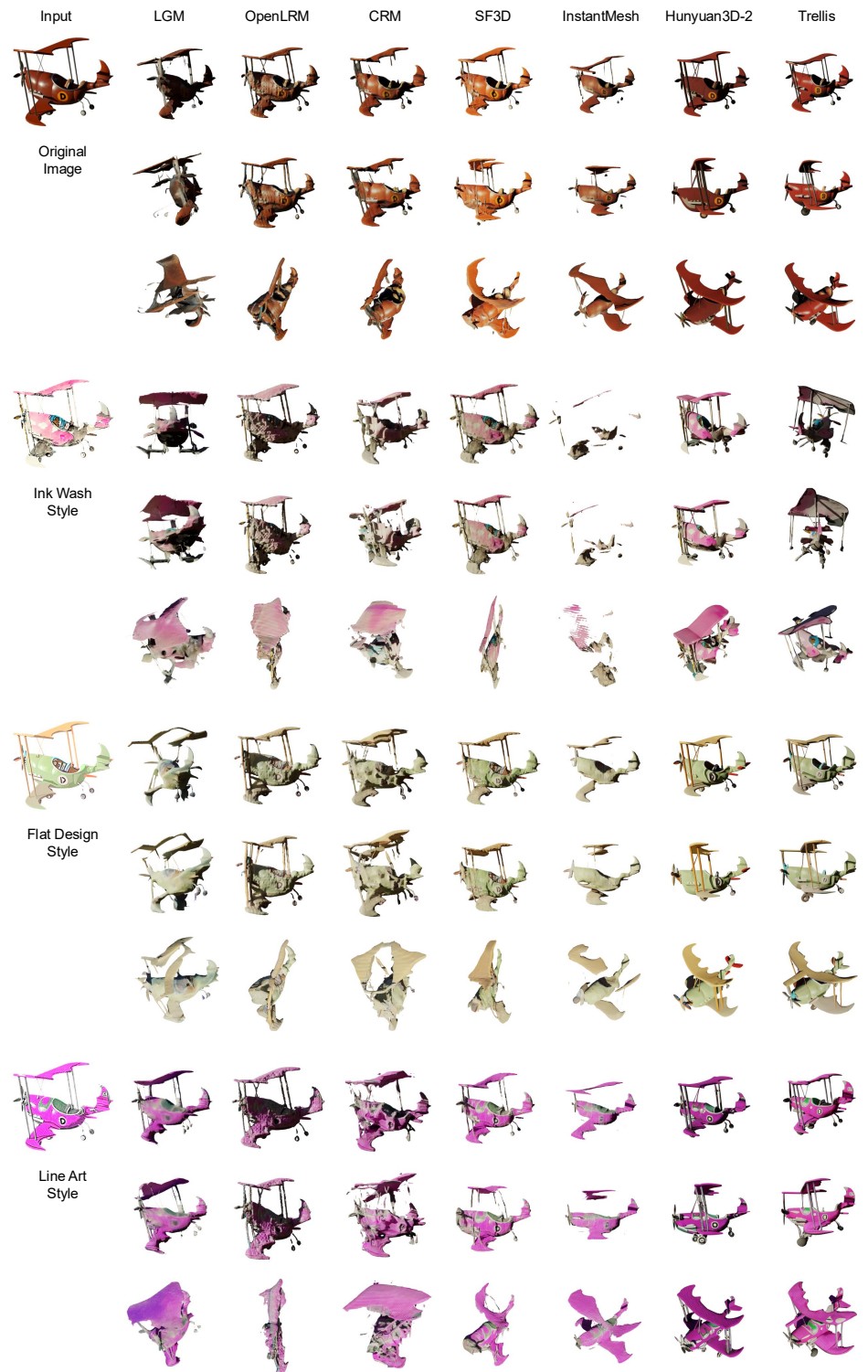

Figure 9: Quantitative results of all perturbations, page 1 / 9.

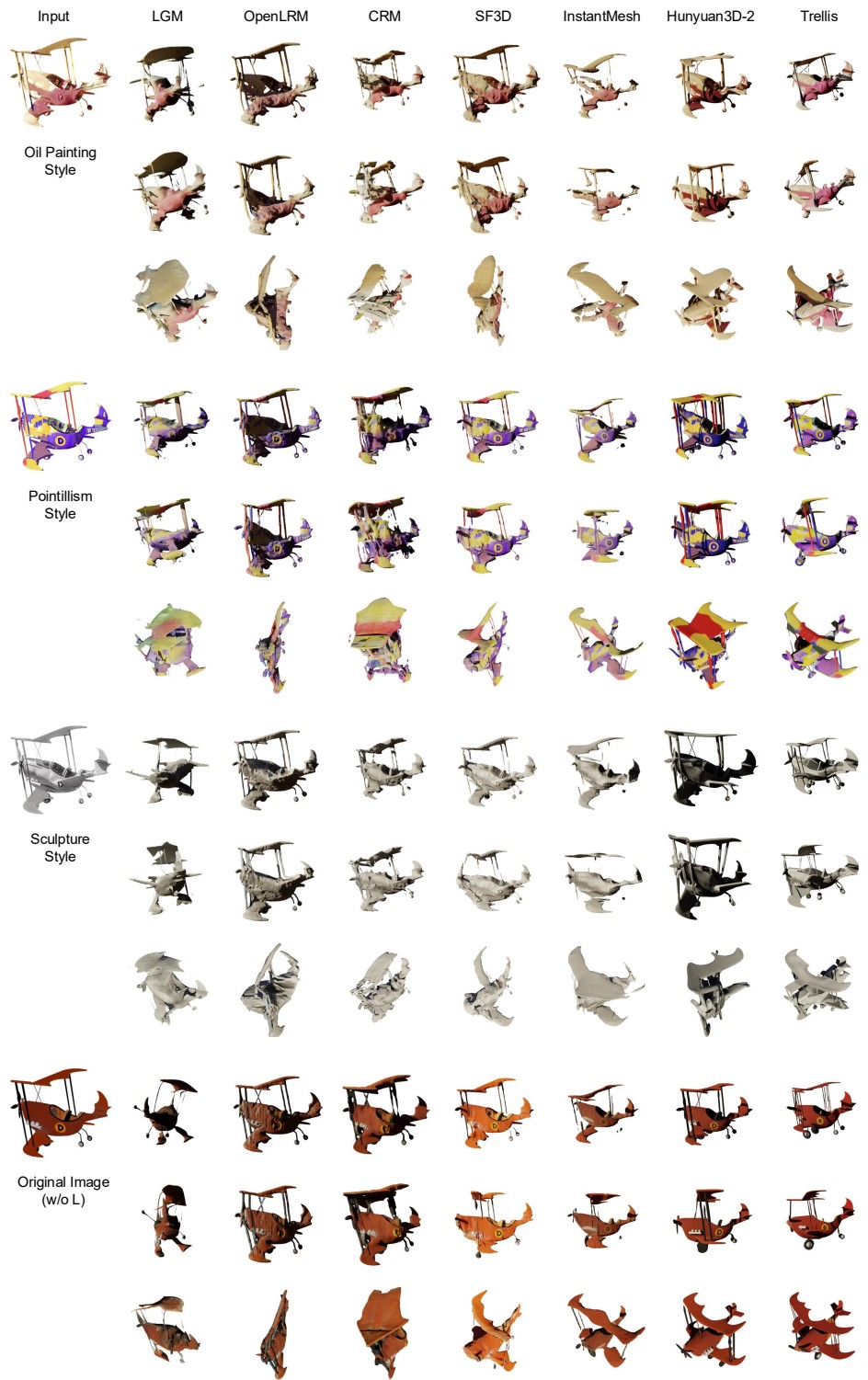

Figure 10: Quantitative results of all perturbations, page 2 / 9.

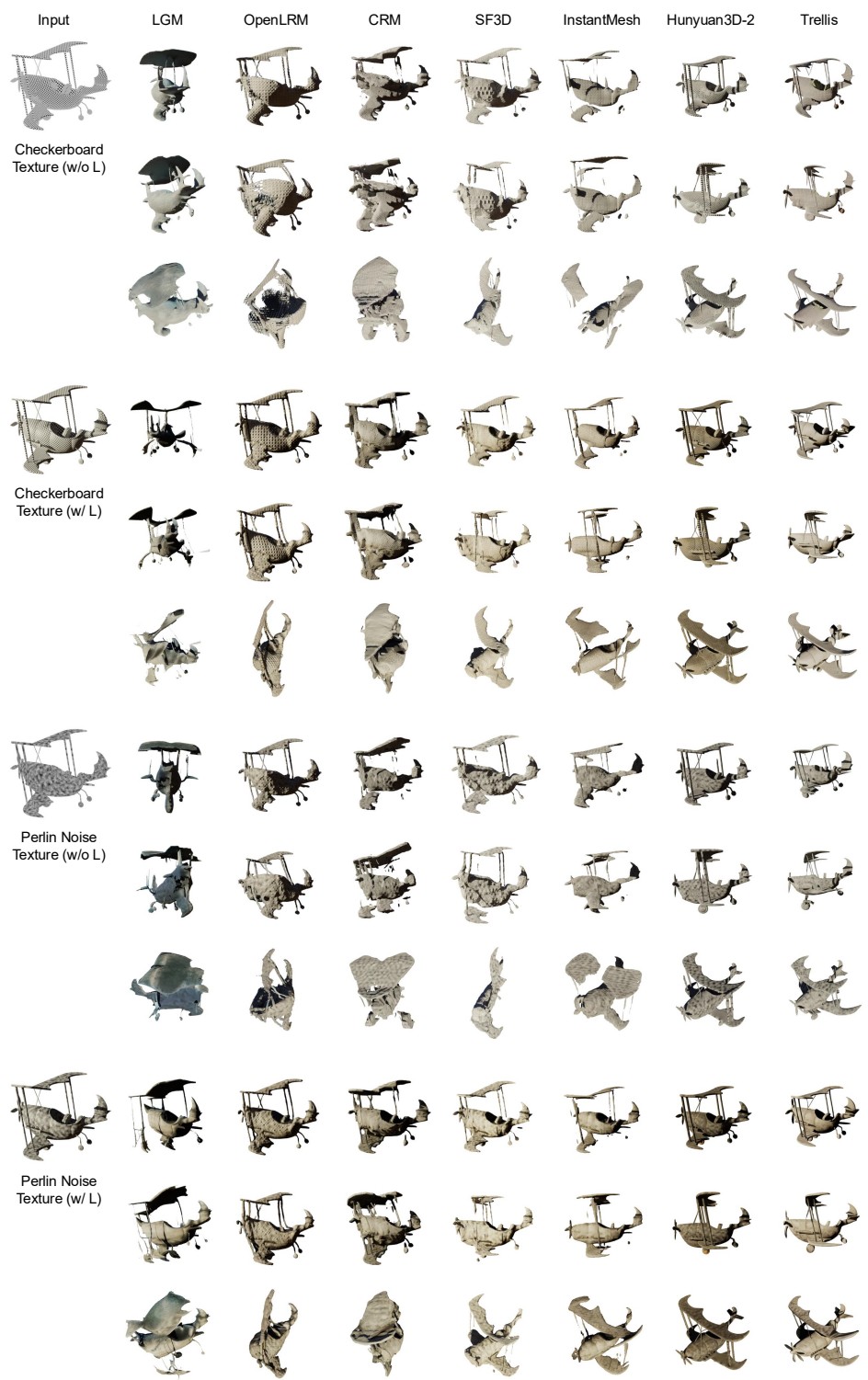

Figure 11: Quantitative results of all perturbations, page 3 / 9.

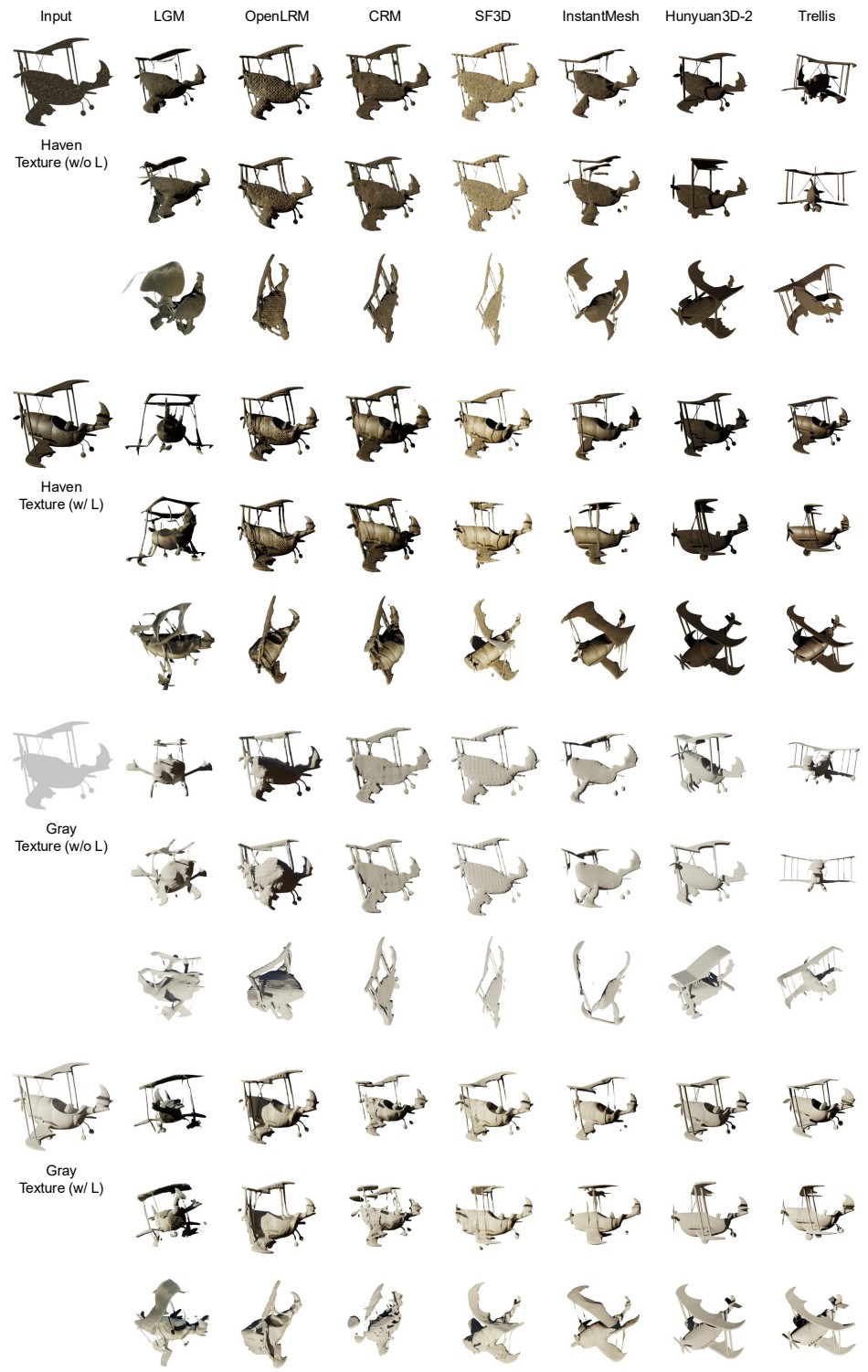

Figure 12: Quantitative results of all perturbations, page 4 / 9.

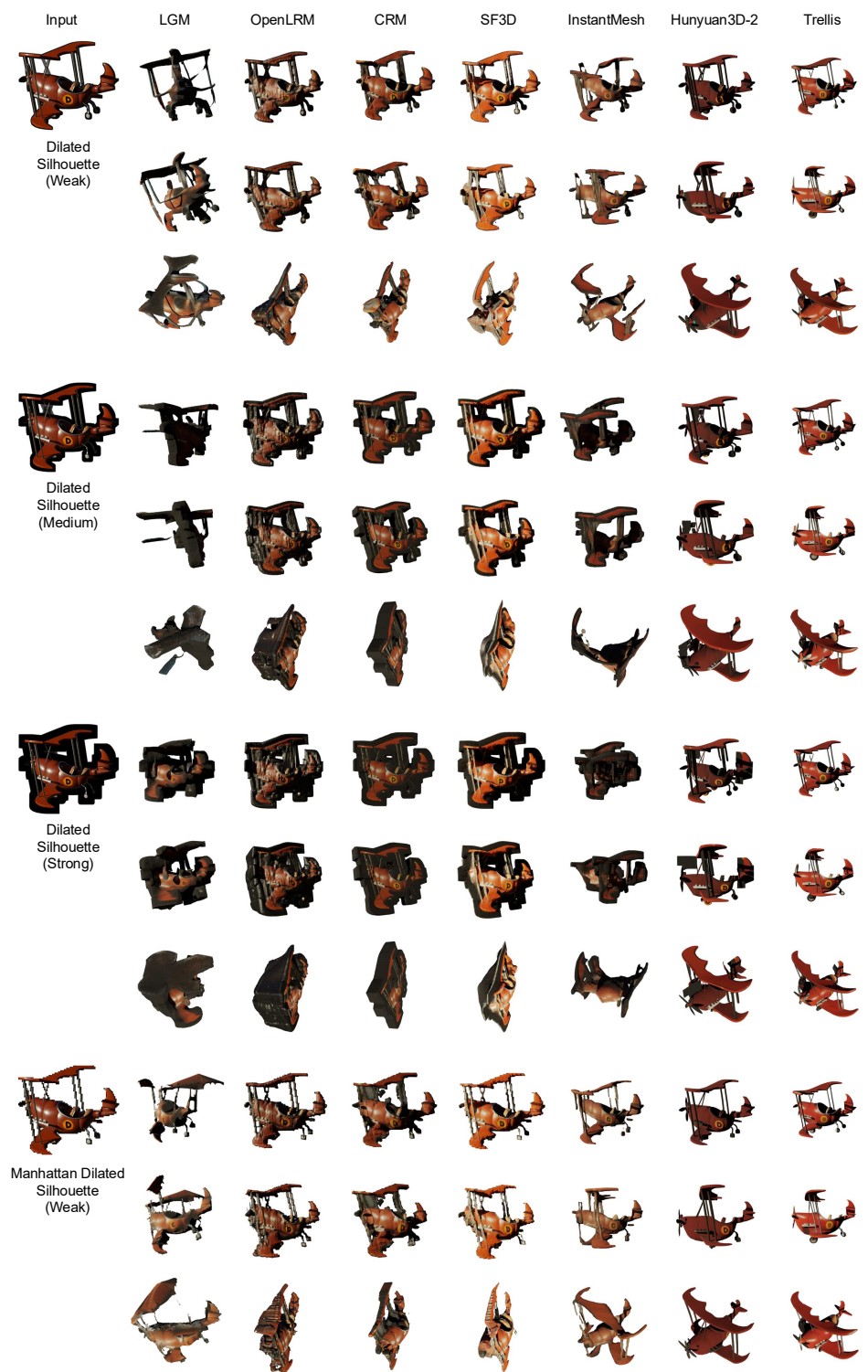

Figure 13: Quantitative results of all perturbations, page 5 / 9.

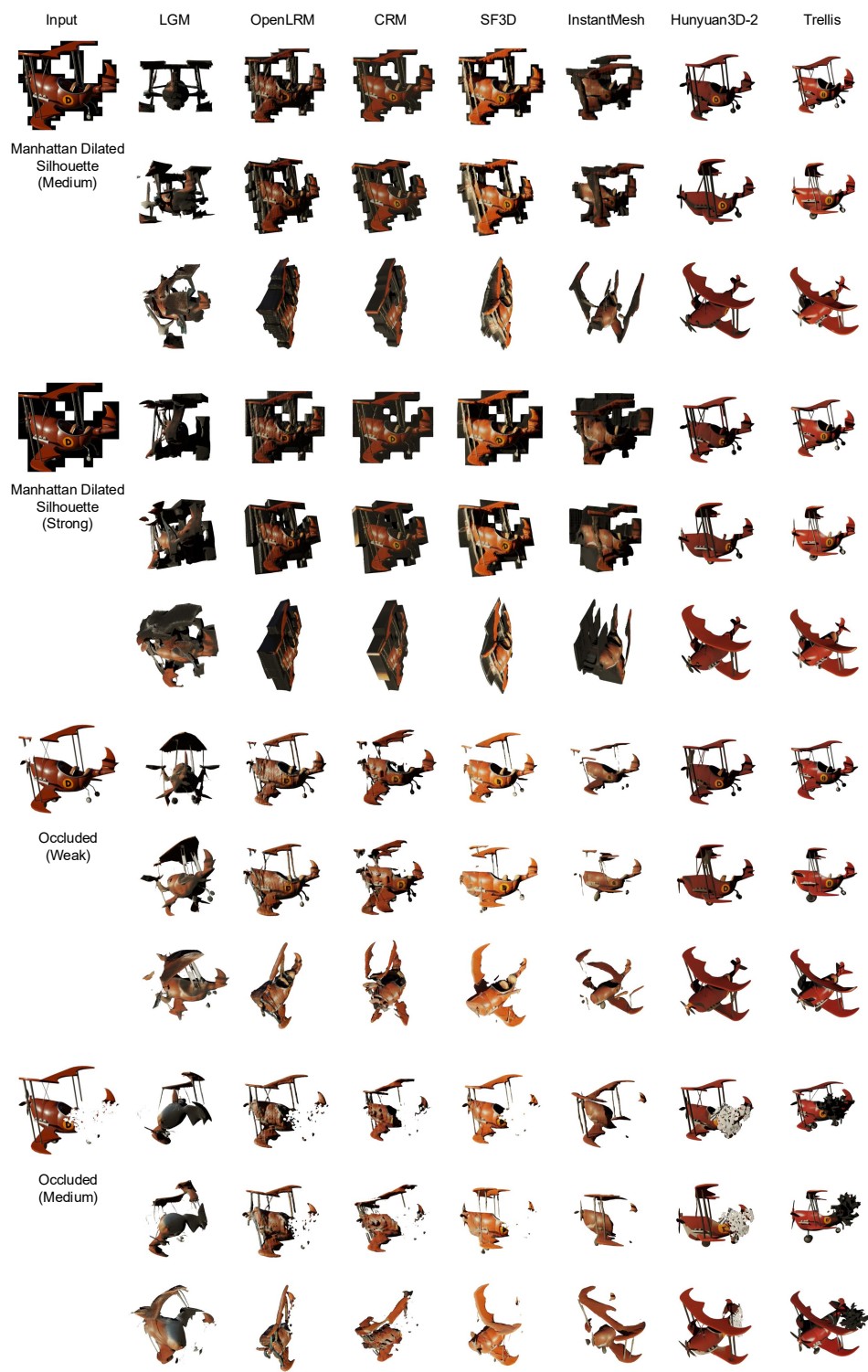

Figure 14: Quantitative results of all perturbations, page 6 / 9.

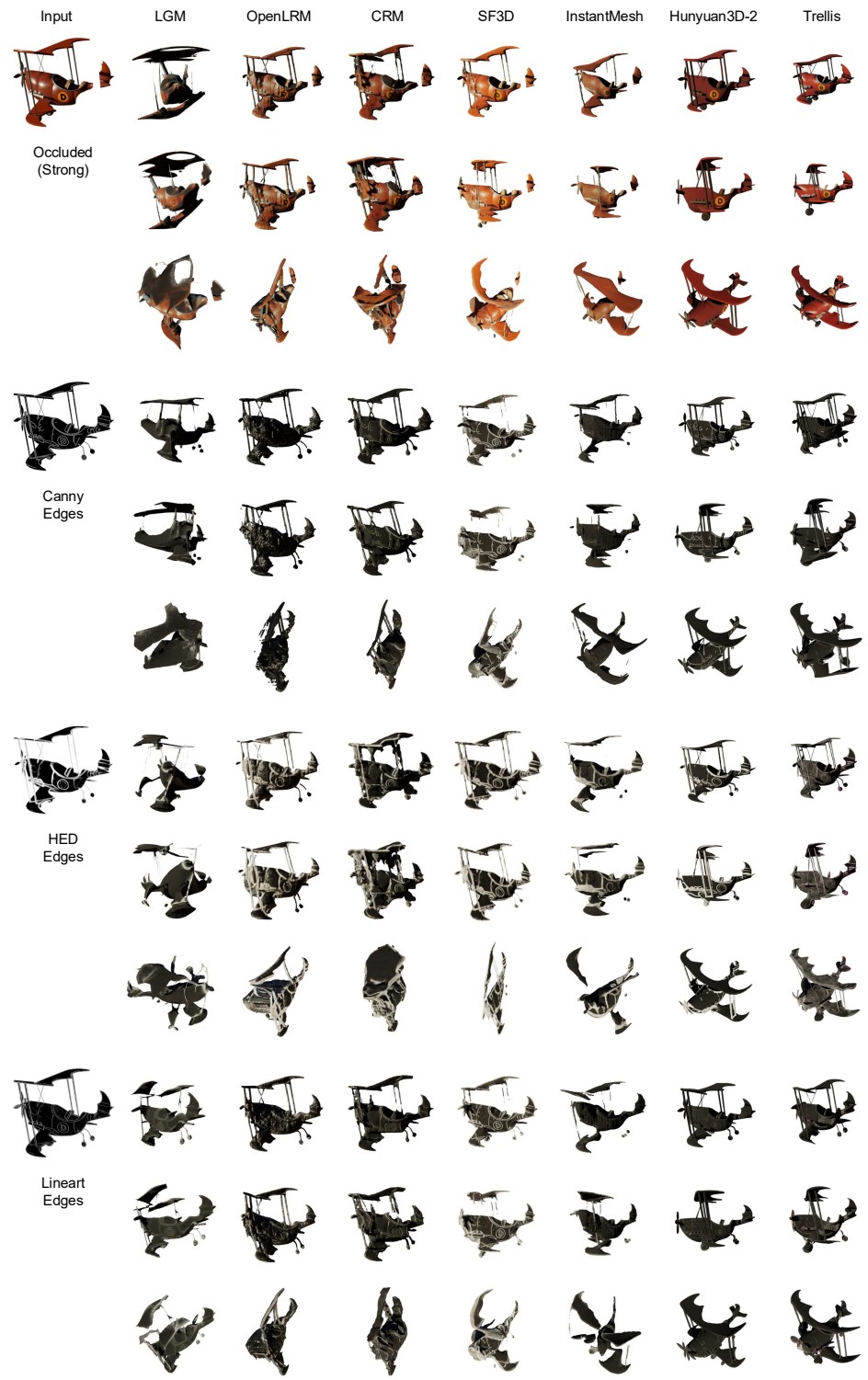

Figure 15: Quantitative results of all perturbations, page 7 / 9.

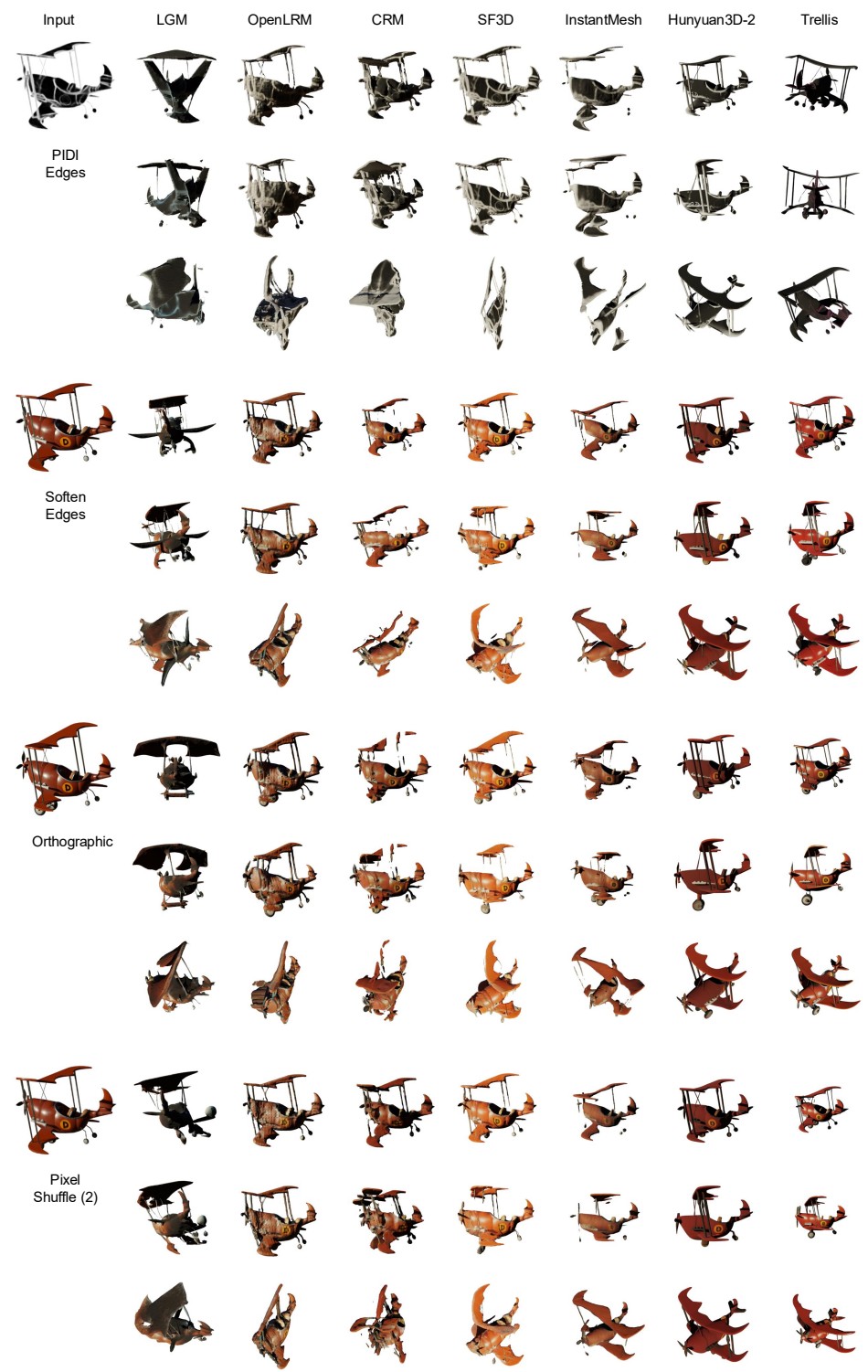

Figure 16: Quantitative results of all perturbations, page 8 / 9.

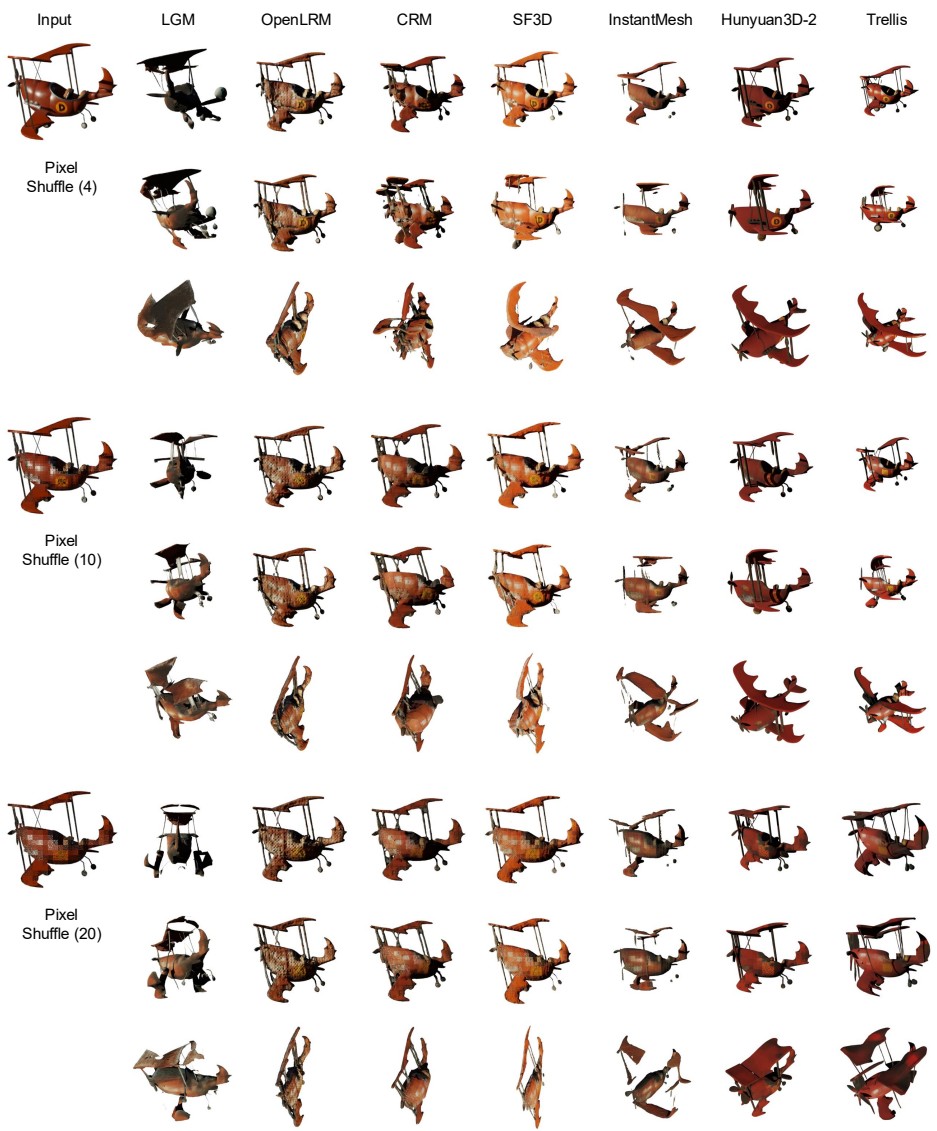

Figure 17: Quantitative results of all perturbations, page 9 / 9.

