# OpenReview forum: "Cue3D: Quantifying the Role of Image Cues in Single-Image 3D Generation"
_NeurIPS.cc/2025/Conference — NeurIPS 2025 spotlight_

### Official Review · Reviewer_ahzB · 2025-06-27

**Clarity:** 3
**Significance:** 3
**Originality:** 3
**Rating:** 5
**Confidence:** 4

**Summary:**

The paper proposes a framework for evaluating the influence of different image cues on image to 3D generation. The framework is based on perturbations of certain cues, and measuring the effect of each pertubation on performance. The authors demonstrate their framework on  a variety of models, trained using different paradigms, and use different metrics to measure the effect of different cues on performance.

**Questions:**

Please see weaknesses.

**Ethical Concerns:**

["NO or VERY MINOR ethics concerns only"]

**Final Justification:**

Thank you to the authors for the detailed response.

Regarding the textual signal: my intention was to highlight that the use of powerful text-to-image models (as base models before fine-tuning) in some of the baselines may contribute to better generalization of meaningful shapes — an aspect I believe is worth discussing.

The authors have addressed most of my concerns, expanded their insightful analysis beyond Zeroverse, organized the quantitative results by paradigm, and added a valuable discussion on how their conclusions can contribute to the training of new image-to-3D models.

Overall, I believe the paper offers valuable insights to the community, and I appreciate the authors’ efforts during the rebuttal. I am happy to raise my score to Accept.

**Limitations:**

Yes.

**Paper Formatting Concerns:**

None.

**Quality:**

2

**Strengths And Weaknesses:**

**Strengths**

1.	The paper proposes a comprehensive analysis of the influence of different image cues on 3D generation. The framework is based on perturbations of certain cues, and evaluation of the effect using different metrics – this framework is interesting and useful.
2.	Image-to-3D is a core and critical task in computer vision, therefore such foundational evaluation/analysis frameworks are valuable to the community and can have a good impact.
3.	The authors inspect a wide range of cues, from style to geometric cues, showing that their framework is generic and capable of providing a range of useful insights.
4.	The authors demonstrate their framework on various image-to-3D models, based on different training techniques, which is crucial for reliable conclusions.
5. The paper is mostly clearly written and easy to follow.

**Weaknesses**

1.	Meaningfulness analysis: the authors found that shape meaningfulness is crucial for high-quality 3D generation results, evaluating the different models of Zeroverse. I have two concerns regarding the meaningfulness evaluation:
*	Zeroverse provides random 3D structures that are semantically meaningless. However, there are other core differences between Zeroverse and datasets like GSO and Toy4K. Most importantly, GSO and Toy4K are based on natural/real-world objects and shapes, which often have internal symmetries that the model can leverage and are relatively "smooth". Zeroverse provide 3D shape that are a random composition of different 3D primitives with augmentations, creating very sharp transitions, unnatural shapes, holes and asymmetric patterns – which are out-of-distribution for the models (and so are the renderings of such objects). None of the models was trained with Zeroverse, hence it is very expected to see a drop in performance due to the out-of-distribution nature of the 3D assets.
* Some of the models used for evaluation are based on pre-trained text-to-image models, trained on huge datasets of text-image pairs. Other models (as Trellis) are trained with text prompts as well. However, the authors completely ignore the text signal used to train some of the models. The text signal may have an important role in the sensitivity of models to meaningful shapes.Additionally, for models that are based on text-conditioned models, it is likely that the base models have not seen prompts that describe shapes as in Zeroverse.
*	To me, it is not clear what would happen if some of the models were fine-tuned with Zeroverse assets (perhaps even with text prompts that describe the 3D structures). It is still possible that the performance would still be underwhelming, reinforcing the authors conclusions, but it is a gap in the evaluation.
2.	How much and **why** different image-to-3D paradigms are sensitive to the different cues: the authors discuss three main paradigms for image-to-3D generation: regression-based, Multiview-based and 3D-based. While the authors show all the results for all the models (specifically the degradation in performance per model per perturbation), one aspect that is lacking analyzing groups of models, trained using one paradigm, and trying to understand whether certain paradigms are more sensitive or more robust to certain cues – and why. For example, following the first point, are methods based on text-to-image models more sensitive to meaningful shape, than regression-based models? The authors do provide some conclusions of that nature in section 4.2, but it feel too hand-wavy and not complete. In table 3, it would have been useful to mention the percentage of degradation in performance rather than the absolute numbers, or to provide a way to compare the sensitivity of different paradigms.
3.	The main motivation of the authors is to provide a comprehensive evaluation framework which can contribute to advancements in image-to-3D generation. While I agree with them about the significance of such works, I think that a more thorough discussion about how the proposed insights can contribute to new image-to-3D models is lacking. Trying to re-train or fine-tune an image-to-3D model using the authors’ conclusions would have been significantly more convincing (I understand it may be out of scope for this research. Yet, a more convincing discussion is important).
4. In the occlusion experiment, the authors propose to dilate the object mask, creating an occluded version of the input image, and mention that many models take object mask as input. The details of this experiment (mostly, the inputs to the model) are not entirely clear to me.
* Are the models in table 3 take object masks as input? To the best of my knowledge, at least some of them were not trained with object mask as input.
* Do the authors only provide the model with an occluded version of the object (without any signal about the occlusion region)? In this case, some "occluded versions" of the images can be a valid image by themselves (as in the example in figure 2 - which to me can be a valid object). Also, the models were never trained to complete occluded parts - so this evaluation setup seems unfair. It would be more convincing to fine-tune a model using occluded objects (so it can learn to complete occluded regions), and then investigate the sensitivity.

---

> ### Author Rebuttal · Authors · 2025-07-31
>
> Thank you for your informative and thoughtful review!
>
> Q1a: Zeroverse domain gap: GSO and Toy4K are based on natural/real-world objects and shapes, with internal symmetries and are relatively "smooth", Zeroverse is not; Model has not see Zeroverse data.
>
> A1a: Thanks for pointing this out. We agree with the reviewer’s observation. We will first describe our motivation of using Zeroverse, and then present an alternative quantitative experiment shape cutmix. This experiment validates that when we (i) reduce the domain gap, (ii) preserve more shape cues like symmetry and smoothness, (iii) on data similar to the training data domain, shape meaningfulness remains an impactful cue.
>
> Our motivation for using zeroverse is because it completely destroys the shape meaningfulness cue. It could be seen as an upper-bound experiment: if the model reasons about 3D shape in a completely geometric way and does not require shape to be meaningful, it should still be able to generate Zeroverse objects. We do agree we should have more experiments on the middle ground: we try to reduce the domain gap and preserve more shape cues when weakening shape meaningfulness.
>
> We introduce several different settings for our CutMix experiments:
> 1. Half-and-half: We mix half of mesh M with the other half of mesh N to construct a new mesh from GSO meshes. This limits the distribution shift and preserves many local and global shape cues (e.g., surface smoothness, local symmetry), and also preserves a significant amount of shape meaningfulness to human perception. We show 3 variants: front-back, left-right, and top-bottom.
> 2. Standard cube-based CutMix. We follow the CutMix paper and randomly sample an axis-aligned 3D cube within the bounding cube of the object. We replace the part of mesh M that falls into the cube with the part from mesh N that falls into the same cube. When sampling the cube, we pin one of its corners at the corner of the object bounding cube to avoid significant discontinuity in the output shape. The length ratio(length_{sampled cube}/length_{bounding cube}) is uniformly sampled from [0.4, 0.6]. Most parts of the object M are outside the chosen cube and remain intact. Meanwhile, the local shape cues are mostly preserved.
>
> 3. OctantMix. We center each mesh and split it into 8 octants by the coordinate planes (xy, yz, and xz planes). Then we replace the part in each octant by the corresponding part from other random meshes from the same dataset. This variant still preserves the local shape cues, and it has a significantly smaller distribution gap than zeroverse compared to our original evaluation data (GSO).
>
> We show the result on the GSO dataset, compared to Zeroverse numbers for reference. We use the CDx1000 metric following Table 3, lower is better.
>
> ||OpenLRM|CRM|SF3D|InstantMesh|Hunyuan3D-2|Trellis|
> |:---|:---|:---|:---|:---|:---|:---|
> |Original|80.89|68.07|61.58|54.54|41.82|39.64|
> |Top-Bottom|93.08(+12.19)|84.05(+15.98)|78.05(+16.47)|80.16(+25.62)|57.94(+16.12)|67.32(+27.68)|
> |Front-Back|88.96(+8.07)|80.79(+12.72)|70.89(+9.31)|69.26(+14.72)|51.74(+9.92)|58.95(+19.31)|
> |Left-Right|90.97(+10.08)|84.44(+16.37)|75.17(+13.59)|80.92(+26.38)|60.56(+18.74)|72.23(+32.59)|
> |CutMix|89.33(+8.44)|78.31(+10.24)|69.77(+8.19)|73.93(+19.39)|59.43(+17.61)|65.36(+25.72)|
> |OctantMix|97.35(+16.46)|89.96(+21.89)|92.43(+30.85)|89.49(+34.95)|75.05(+33.23)|79.94(+40.30)|
> |Zeroverse|96.59(+15.70)|81.45(+13.38)|90.34(+28.76)|89.47(+34.93)|78.09(+36.27)|78.14(+38.50)|
>
> The results show that all variants still significantly harms performance. In particular, despite a significantly smaller domain gap to GSO, OctantMix leads to a similar performance drop as zeroverse, indicating catastrophic failure. Standard CutMix only alters approximately 1/8 of the mesh volume, but it also causes a very significant drop(e.g., 20 points for Hunyuan3D-2). Even for the minimal half-and-half perturbations, where shape meaningfulness still largely remains, we still observe significant performance drop of >10 points most of the time. This performance drop is still very significant compared to the effect of other cues. These results highlight the importance of shape meaningfulness, since the models have significant difficulty to generalize even when we only partially perturb this cue, especially compared to other cues like texture meaningfulness.
>
>
>
>
> Q1b: Some models are based on text-to-image models, some are trained on text prompts. Text signal corresponding to Zeroverse is unseen.
>
> A1b: We appreciate your comment and want to clarify a potential misunderstanding: we did not use text as input in our experiments, and none of the models have directly seen paired text and 3D assets. Regarding Trellis specifically, we used its image-to-3D checkpoint, which indeed was not trained on any text. You are correct that Multi-view diffusion (MVD)-based models (LGM, CRM, InstantMesh) starts from text-to-image models before further training on Objaverse, while the other models haven't seen any text during training. While text-based pretraining might influence the capabilities of the three MVD-based models, we believe this is not a major concern because both text-pretrained models and those without text pretraining exhibited similarly significant performance drops under shape meaningfulness perturbation. Since our main focus is on image-to-3D modeling, we believe exploring text-based generalization would be valuable future work.
>
>
>
> Q2: Paradigm-wise sensitivity to cues and why; Percentage of degradation
> A2: Thank you very much for this helpful suggestion. We completely agree that grouping the models by paradigm and quantifying their relative sensitivity helps clarify which cues each family relies on most. Below, we summarize our current results paradigm-wise using percentages and provide additional interpretation.
> The following table shows the percentage of performance change on the GSO dataset. Since directly computing percentages on Chamfer Distance is not meaningful, we first calculate an F1 score at a Chamfer Distance threshold following SF3D [1], and then compute the percentage change based on this F1 score:
>
> ||Regression-based|Multi-view|Native3D|
> |:--|:--|:--|:--|
> |Shading|-7.78%|-3.54%|-6.77%|
> |Silhouette|-21.41%|-14.62%|-4.81%|
> |Occlusion|-5.95%|-10.76%|-11.69%|
> |Edges|-5.76%|-7.24%|-6.71%|
> |Perspective|-8.28%|-3.23%|-5.66%|
> |Style|-7.88%|-6.17%|-11.64%|
> |Local continuity|-8.92%|-8.20%|-12.02%|
>
> It's worth noting that no metric is perfectly linear with respect to model performance, so direct percentage comparisons might not be entirely fair between higher- and lower-performing models. Nonetheless, we can still draw several interesting conclusions. For instance, multi-view models seem less affected by shading due to their rich pretraining on text-to-image data. Native 3D models appear more robust to silhouette perturbations, possibly due to their generative training objectives. We'll include a more detailed discussion of these insights in the final version of the paper.
>
>
>
> Q3: More discussion on how proposed insights can contribute to new image-to-3D models.
>
> A3: We sincerely appreciate this question, and we fully understand your intention to help improve our paper. We agree that illustrating how our conclusions can inspire future research is valuable. Due to limited time and resources, we were not able to retrain or fine-tune multiple models. However, we made an effort to illustrate how our insights could inspire new research directions. Specifically, we explored a line-art-to-3D application, where the input is a simplified line-art image (in our case, extracted from GSO images), and the goal is to recover the underlying 3D structure. Below, we report results in terms of Chamfer Distance x 1000 following Table 3 (lower values indicate better performance):
>
>
> |Input Image|InstantMesh|SF3D|Trellis|
> |:---|:---|:---|:---|
> |Original|54.5|61.6|39.6|
> |Line-art|66.1|79.5|51.3|
> |Line-art + Geometric Cues|59.4|67.4|50.0|
>
> Our Cue3D analysis emphasizes that semantics alone are not sufficient and highlights the importance of geometric cues (L57). This point is especially relevant in the line-art-to-3D scenario, as line-art inherently lacks detailed surface properties to inform 3D reconstruction. Directly using line-art images results in significantly worse 3D reconstructions than original images. Inspired by our analysis, we leveraged an image diffusion model (Flux ControlNet) conditioned on line-art, prompting it to generate geometric cues (e.g., shading and texture) through prompting for 3D rendering-style outputs. This approach significantly improved performance, validating that our proposed insights could meaningfully contribute to future research in image-to-3D.
>
>
> Q4: Occlusion experiment clarification; Occluded version could be a valid object; Fine-tuning a model using occluded objects.
>
> A4:  Thanks for bringing up this important point. To clarify, all models evaluated take masks as input and have been trained specifically on image+mask combinations. Your suggestion that occluded versions of objects could sometimes represent valid objects is indeed insightful, as occlusion naturally introduces ambiguity. However, our primary objective is to evaluate whether the models have learned a robust prior distribution of plausible 3D shapes. Ideally, models should reconstruct shapes that align with their learned 3D priors, despite ambiguity from occlusions. Additionally, we fully acknowledge that fine-tuning on occluded images could further improve performance. To explore this, we're currently evaluating Amodal3R [2], a version of Trellis fine-tuned explicitly on occluded images, on our occlusion perturbation. We will include these results as soon as the evaluation is complete.
>
> Reference
>
> [1] Boss et. al., SF3D: Stable Fast 3D Mesh Reconstruction with UV-unwrapping and Illumination Disentanglement, 2025
>
> [2] Wu et. al., Amodal3R: Amodal 3D Reconstruction from Occluded 2D Images, 2025

---

> > ### Comment · Reviewer_ahzB · 2025-08-01
> >
> > Thank you to the authors for the detailed response.
> > Regarding the textual signal: my intention was to highlight that the use of powerful text-to-image models (as base models before fine-tuning) in some of the baselines may contribute to better generalization of meaningful shapes — an aspect I believe is worth discussing.
> > The authors have addressed most of my concerns, expanded their insightful analysis beyond Zeroverse, organized the quantitative results by paradigm, and added a valuable discussion on how their conclusions can contribute to the training of new image-to-3D models.
> > Overall, I believe the paper offers valuable insights to the community, and I appreciate the authors’ efforts during the rebuttal. I am happy to raise my score to Accept.

---

> > > ### Author Response · Authors · 2025-08-01
> > >
> > > Thank you very much for your insightful comments and for raising your score! We fully agree with your point regarding the impact of text-to-image base models on shape generalization, and we will incorporate this discussion from our rebuttal into the main paper.

---

### Official Review · Reviewer_bRRe · 2025-07-02

**Clarity:** 3
**Significance:** 4
**Originality:** 3
**Rating:** 5
**Confidence:** 4

**Summary:**

The paper introduces Cue3D, a novel framework designed to evaluate the influence of individual image cues on single-image 3D generation. Its core idea is to systematically analyze how different image cues, such as shading, texture, and silhouette, affect the performance of various single-image 3D generation methods. The main contributions of the paper include the creation of a unified benchmark for evaluating seven state-of-the-art methods across different paradigms, and the proposal of a perturbation analysis to measure the significance of each image cue. The paper reveals that shape meaningfulness and geometric cues, especially shading, are crucial for 3D generation, and that models may overly rely on provided silhouettes. This work is significant to the academic community as it enhances the understanding of how modern 3D networks utilize classical vision cues, offering guidance for developing more transparent, robust, and controllable single-image 3D generation models and bridging the gap between classical computer vision and deep learning approaches.

**Questions:**

1. You've focused on several key image cues. Are there any other image cues that might also have a significant impact on single-image 3D generation, but were not considered in this study?

2. How generalizable are the findings of this study across different datasets and categories of objects? Have you conducted any experiments to test the robustness of these conclusions?

**Ethical Concerns:**

["NO or VERY MINOR ethics concerns only"]

**Final Justification:**

The authors basically addressed my concerns in the rebuttal. I believe the findings are significant to the academic community as it enhances the understanding of how modern 3D networks utilize classical vision cues, offering guidance for developing more transparent, robust, and controllable single-image 3D generation models. I remain postive about the paper.

**Limitations:**

yes

**Paper Formatting Concerns:**

I do not notice any major formatting issues

**Quality:**

3

**Strengths And Weaknesses:**

Strengths
1. Cue3D is the first comprehensive and model-agnostic framework for quantifying the influence of individual image cues in single-image 3D generation. This is a novel and important contribution to the field as it addresses the gap in understanding how modern 3D networks leverage classical vision cues. Second,
2. The paper presents a unified benchmark covering seven state-of-the-art methods across different paradigms, providing a systematic way to evaluate and compare these methods. The perturbation analysis conducted in the paper offers valuable insights into the dependencies of single-image 3D models on various image cues, which can guide future research in developing more transparent, robust, and controllable models.

Weaknesses
While the paper provides a detailed analysis of the role of different image cues, it does not explore how these cues might interact with each other in more complex ways.

---

> ### Author Rebuttal · Authors · 2025-07-31
>
> Response to Reviewer bRRe:
>
> Thank you so much for your thoughtful and helpful comments!
>
> Q1: How do the cues interact with each other in more complex ways?
>
> A1: This is a really interesting point. The main difficulty in exploring more complex interactions between cues is that certain cues(e.g., texture and style) are inherently difficult to be combined in a meaningful way. We initially selected these cues because of their perceptual importance and interpretability to humans, rather than strict orthogonality. However, we completely agree that exploring these more nuanced interactions is a great direction for future research!
>
> Q2: Are there other important cues?
>
> A2: Great question! Yes, there are other important cues worth investigating. For example, as Reviewer MKBa also mentioned, material properties (like specularity and roughness), variations in lighting conditions, and viewing angles could all provide interesting insights.
> We added an initial experiment exploring the interaction between shading and material properties (e.g., specularity). Specifically, we conduct this additional experiment on the GSO dataset and report results in CDx1000 following Table 3 (lower is better). We investigated combinations of environment map lighting vs. directional lighting, and default vs. large specularity.
>
> ||OpenLRM|CRM|SF3D|InstantMesh|Hunyuan3D-2|Trellis|
> |:---|:---|:---|:---|:---|:---|:---|
> |env map w/o large specularity(original)|80.89|68.07|61.58|54.54|41.82|39.64|
> |env map w/ large specularity|81.70(+0.81)|68.40(+0.33)|61.84(+0.26)|54.43(-0.11)|42.32(+0.50)|38.94(-0.70)|
> |directional w/o large specularity|84.19(+3.30)|72.00(+3.93)|67.26(+5.68)|60.60(+6.06)|42.77(+0.95)|43.97(+4.33)|
> |directional w/ large specularity|84.47(+3.58)|70.68(+2.61)|65.99(+4.41)|57.94(+3.40)|42.76(+0.94)|42.54(+2.90)|
>
> This experiment yields interesting observations! Surprisingly, directional lighting significantly degraded performance for most models, except Hunyuan3D-2. Additionally, large specularity slightly improved performance in several cases, although not always, with the exception of OpenLRM and Hunyuan3D-2. We speculate that this might be related to the different lighting setup used in each model’s training.
>
> One major benefit of Cue3D is that it is easily extensible. We expect more cues like this could be integrated in the future into our Cue3D framework.
>
> Q3: How generalizable are the findings? Test the robustness of the conclusion.
>
> A3: Thank you for raising this important point! We are quite confident that our findings are generalizable, since our findings generalizes across scanned real-world object dataset (GSO) and synthetic 3D assets dataset (Toys4K). We will also include a figure that analyzes per-category performance impact on Toys4K in the final version of the paper. Our conclusions are consistent across these different datasets and categories. Regarding the robustness, we include Appendix Table 4 to illustrate that our conclusions are robust to random seed. Meanwhile, also in response to Reviewer wyxx Q2, we will include statistical tests(specifically, Wilcoxon signed-rank tests) to clearly demonstrate that the observed drops in performance metrics are statistically significant.
>
> We sincerely appreciate your valuable review. If you have any further comments or questions, we would be happy to discuss further!

---

> > ### Comment · Reviewer_bRRe · 2025-08-04
> >
> > Thanks to the authors for the detailed response. The authors addressed my concerns on generalization of the findings of this study across different datasets. I will remain the Accept rating.

---

### Official Review · Reviewer_MKBa · 2025-07-02

**Clarity:** 3
**Significance:** 2
**Originality:** 3
**Rating:** 5
**Confidence:** 3

**Summary:**

This paper presents Cue3D, a way to analyse (quantitavely) how much single-image
to 3D methods rely on different kinds of information in the input image, and applies
it to various existing single-image to 3D methods. The focus is on both geometry and
appearance, and on the object-centric task (i.e. not single-image to 3D scene, although
much of the methodology could carry over).

**Questions:**

If I understand correctly, "Shape Meaningfullness" is unlike the other cues in that
instead of modifying the input to alter it in some way, here the input is simply replaced
by one consisting of randomly combined shapes. Since difficulty of reconstruction naturally
various across examples (and presumably depends also on complexity as well as the other
cues discussed here), how do we ensure that this is a fair comparison – i.e. that the
non-meaningful shapes are in some sense of "similar" difficulty to the ones
they are replacing, and are textured and shaded in a similar way? Without this, I don't
see how to make a meaningful comparison between geometry quality on Zeroverse and the
other datasets.

Can we be confident that the methods being analysed have not seen Toys4K or GSO data
(or indeed Zeroverse) during training?

**Ethical Concerns:**

["NO or VERY MINOR ethics concerns only"]

**Final Justification:**

My questions were answered very satisfactorily. After reading the other reviews and discussion, including around the new CutMix experiments, I upgrade my rating to Accept.

**Limitations:**

There is a limitations section in the Appendix. I would suggest moving it to the main
paper and making it more detailed. It talks about considering a wider range of models
and datasets, but there are also limitations in the choice of cues and how they are
implemented for evaluation.

**Paper Formatting Concerns:**

There are many capitalization errors in the titles of papers in the References section.
(Examples: "3d" for "3D", "cnns" for "CNNs", "bayes" for "Bayes", "Shap-e" for "Shap-E",
etc, etc)

**Quality:**

2

**Strengths And Weaknesses:**

[Strength] The authors identify a reasonable set of different image properties (shading,
texture, etc) to investigate, and apply sensible metrics to quantify both geometry
(via chamfer distance, and with a sensible distinction between seen and unseen geometry)
and appearance (LPIPS, PSNR etc).

[Strength] The paper is clearly written and it's easy to understand the methodology.
The supplementary material gives plenty of examples to illustrate the kinds of input
variation considered, and the results obtained.

[Both] The general conclusions (nicely summarized in line 48–75) don't seem
particularly surprising. On the other hand, the data here can highlight some
differences between approachs which I think is valuable.

[Weakness] In some cases it's not clear (to me, at least) that the input variations
are meaningful. Applying style-transfer to the input certainly does _something_ but
going by the examples in the appendix, "Pointillism style", "Oil Painting Style"
and "Line Art Style" don't look at all like the styles they are supposed to me. In
other cases, it's easy to think of further transforms that would be interesting.
For example, having discovered the important of "shading", one could generate
inputs using various different types of lighting and changing material properties
such as specularity and roughness, to try to understand specifically _what_ aspects
of shading are crucial. E.g. directional lighting without specularity, directional
lighting with specularity, multiple light sources, environment map, self-shadowing,
etc. Another natural question is: how does the 3D quality vary with the viewing
angle of the input image (side view vs top-down view).

[Strength] The authors promise to release the code on acceptance, and use only public
datasets. This should make it possible to others to use Cue3 to evaluate new methods,
and to extend it.

---

> ### Author Rebuttal · Authors · 2025-07-31
>
> Thank you for your insightful comments!
>
> Q1: Whether style variations are meaningful.
>
> A1: We appreciate your thoughtful observation! We used a state-of-the-art style transfer model, but we acknowledge it is not perfect. While it successfully applies the desired style in many cases, there are scenarios where it does not fully capture the reference style. However, our goal for using style transfer is to perturb geometric cues while preserving semantic content (L175). Even when the style transfer is not perfect, it still generally achieves our goal of introducing meaningful perturbations.
>
> Q2: More cues and transforms; Effect of varying viewing angle.
>
>
> A2: Thanks you for the suggestions! Given the limited time and resources during the rebuttal period, we prioritized performing a more in-depth experiment exploring the interaction between shading and material properties (e.g., specularity). Specifically, we conducted this additional experiment on the GSO dataset and report results in CDx1000 following Table 3 (lower is better). We investigated combinations of environment map lighting vs. directional lighting, and default vs. large specularity.
>
> ||OpenLRM|CRM|SF3D|InstantMesh|Hunyuan3D-2|Trellis|
> |:---|:---|:---|:---|:---|:---|:---|
> |env map w/o large specularity(original)|80.89|68.07|61.58|54.54|41.82|39.64|
> |env map w/ large specularity|81.70(+0.81)|68.40(+0.33)|61.84(+0.26)|54.43(-0.11)|42.32(+0.50)|38.94(-0.70)|
> |directional w/o large specularity|84.19(+3.30)|72.00(+3.93)|67.26(+5.68)|60.60(+6.06)|42.77(+0.95)|43.97(+4.33)|
> |directional w/ large specularity|84.47(+3.58)|70.68(+2.61)|65.99(+4.41)|57.94(+3.40)|42.76(+0.94)|42.54(+2.90)|
>
> This experiment yields interesting observations! Surprisingly, directional lighting significantly degraded performance for most models, except Hunyuan3D-2. Additionally, large specularity slightly improved performance in several cases, although not always, with the exception of OpenLRM and Hunyuan3D-2. We speculate that this might be related to difference lighting setup used in each model’s training. Meanwhile, investigating the effect of varying viewing angles would be valuable, and we believe this is a promising direction for future research.
>
> Q3: Zeroverse comparison may be unfair due to difficulty.
>
> A3: We appreciate the reviewer’s observation. We additionally conducted several shape CutMix experiments, which mixes different assets from the GSO dataset to create new meshes, with the objective of partially removing shape meaningfulness while maintaining a similar difficulty. We introduce several different settings for our CutMix experiments:
> 1. Half-and-half: We mix half of mesh M with the other half of mesh N to construct a new mesh from GSO meshes. This limits the distribution shift and preserves many local and global shape cues (e.g., surface smoothness, local symmetry), and also preserves a significant amount of shape meaningfulness to human perception. We show 3 variants: front-back, left-right, and top-bottom.
> 2. Standard cube-based CutMix. We follow the CutMix paper and randomly sample an axis-aligned 3D cube within the bounding cube of the object. We replace the part of mesh M that falls into the cube with the part from mesh N that falls into the same cube. When sampling the cube, we pin one of its corners at the corner of the object bounding cube to avoid significant discontinuity in the output shape. The length ratio(length_{sampled cube}/length_{bounding cube}) is uniformly sampled from [0.4, 0.6]. Most parts of the object M are outside the chosen cube and remain intact. Meanwhile, the local shape cues are mostly preserved.
>
> 3. OctantMix. We center each mesh and split it into 8 octants by the coordinate planes (xy, yz, and xz planes). Then we replace the part in each octant by the corresponding part from other random meshes from the same dataset. This variant still preserves the local shape cues, and it has a significantly smaller distribution gap than zeroverse compared to our original evaluation data (GSO).
>
> We show the result on the GSO dataset, compared to Zeroverse numbers for reference. We use the CDx1000 metric following Table 3, lower is better.
>
> ||OpenLRM|CRM|SF3D|InstantMesh|Hunyuan3D-2|Trellis|
> |:---|:---|:---|:---|:---|:---|:---|
> |Original|80.89|68.07|61.58|54.54|41.82|39.64|
> |Top-Bottom|93.08(+12.19)|84.05(+15.98)|78.05(+16.47)|80.16(+25.62)|57.94(+16.12)|67.32(+27.68)|
> |Front-Back|88.96(+8.07)|80.79(+12.72)|70.89(+9.31)|69.26(+14.72)|51.74(+9.92)|58.95(+19.31)|
> |Left-Right|90.97(+10.08)|84.44(+16.37)|75.17(+13.59)|80.92(+26.38)|60.56(+18.74)|72.23(+32.59)|
> |CutMix|89.33(+8.44)|78.31(+10.24)|69.77(+8.19)|73.93(+19.39)|59.43(+17.61)|65.36(+25.72)|
> |OctantMix|97.35(+16.46)|89.96(+21.89)|92.43(+30.85)|89.49(+34.95)|75.05(+33.23)|79.94(+40.30)|
> |Zeroverse|96.59(+15.70)|81.45(+13.38)|90.34(+28.76)|89.47(+34.93)|78.09(+36.27)|78.14(+38.50)|
>
> The results show that all variants still significantly harms performance. In particular, despite a significantly smaller domain gap and a more similar difficulty to GSO, OctantMix leads to a similar performance drop as zeroverse, indicating catastrophic failure. Standard CutMix only alters approximately 1/8 of the mesh volume, but it also causes a very significant drop(e.g., 20 points for Hunyuan3D-2). Even for the minimal half-and-half perturbations, where shape meaningfulness still largely remains, we still observe significant performance drop of >10 points most of the time. This performance drop is still very significant compared to the effect of other cues. These results highlight the importance of shape meaningfulness, since the models have significant difficulty to generalize even when we only partially perturb this cue, especially compared to other cues like texture meaningfulness.
>
> Q4: Can we be confident that test sets are not seen in training.
>
> A4: This is a valid and important question. We are quite confident that none of the methods have seen the GSO or Zeroverse datasets during training, as GSO is widely reserved as a test-only benchmark, and Zeroverse is also not typically used for training single-image-to-3D models. We are fairly confident that Toys4K is not seen by all the models(Page 6 of the trellis paper: “For quantitative evaluations, we use Toys4k, which is not part of our training set or those of the compared methods.”), except that we are not absolutely certain about Hunyuan3D-2, since they do not release or list their training data. Since Hunyuan3D-2 does not behave significantly differently from trellis on Toys4K, it is likely that they also did not use Toys4K in training.
>
> Q5: Limitation section and formatting.
>
> A5: Thank you for the suggestion! We will move the limitation section to the main paper and make it more comprehensive in the final version. We will also fix the reference formatting issues.
>
> Thank you once again for your insightful and helpful feedback. Please feel free to share any additional suggestions or questions you may have!

---

> > ### Comment · Reviewer_MKBa · 2025-08-04
> >
> > Thank you for the response, which clearly represents a substantial amount of work and experimentation.
> >
> > I think your new results do strengthen the paper (although a fuller investigation of the effect of lighting and shading is no doubt warranted in future, as this seems to be the most crucial area). The CutMix experiments satisfy my concerns about Zeroverse, but I see that reviewer wyxx also had concerns about shape meaningfulness, so I will discuss this with them before deciding on my final rating.

---

### Official Review · Reviewer_wyxx · 2025-07-02

**Clarity:** 2
**Significance:** 2
**Originality:** 3
**Rating:** 5
**Confidence:** 4

**Summary:**

The paper seeks to gain insights into how recent image-to-3D object reconstruction methods reason about their inputs to generate their output. Thus, it seeks to impose greater interpretability on these methods, which all learn complex latent representations and often function as black-boxes. It does this by evaluating the influence of different cues on the quality of 3D reconstruction as measured by several well-established metrics. The authors conclude that meaningful shapes, shading, occlusion and silhouettes are among the most important cues used by these methods.

**Questions:**

1. (Evaluation) The results in Table 3 and 5 seem to be fundamental in supporting the main claims of the paper. However, the presentation format of these results makes them very unreadable, and difficult to derive any insights from. I would suggest using more illustrative graphical examples to present these results. If space is an issue, I would suggest focusing less on the performance comparison of each method (Sec. 4.1) since that doesn't seem to be primary to the objective of the paper.

Action: The authors should provide a description -- preferably a list -- of the different visualizations they plan to use to better illustrate the data in Table 3 and 5. A good example of a paper that does this well is "VBench: Comprehensive Benchmark Suite for Video Generative Models."

2. (Evaluation) In an evaluation-based work like this, it can be useful to provide some statistical tests to show that the drop in metrics are actually significant.

3. (Evaluation) How is the performance robustness of each cue determined in Fig.1 (right)?

4. (Methodology) The authors' claim that shape meaningfullness is the most important factor in determining model performance seems to be based on a flawed criterion: it evaluates models on a shape dataset that is outside its training distribution. But any learned model will fail when the testing and training distributions don't match. How do the authors' determine that the drop in performance in this case is because the shapes aren't meaningful (with the term "meaningful" being defined by the training data of the model)? A better strategy to evaluate this cue may be to use something similar to a cut-mix augmentation on shapes from the datasets that the models have been trained on, rather than using the entirely out-of-distribution Zeroverse dataset for evaluation.

Action: The authors should clarify how they disentangle the effects of model-inherent shape "meaningfulness" from a simple failure to generalize to out-of-distribution samples (which any model with limited capacity will suffer from). I will be entirely convinced if the authors can show that the cut-mix augmentation causes a similar degradation of performance on the shape-meaningfulness cue.

5. (Methodology) How do the authors ensure that each individual cue is entirely isolated? For instance, the stylization and texturing cues can interfere with one another, as can the edge cue.

Action: A response from the authors about how they ensure the impact of cues is isolated would help clarify this point. Ideally, it would be wonderful to have more detailed statistical analysis in the paper that can directly yield insights into such questions. For instance, running PCA on the (6x9x3x3x2x1x4) dimensional tensor that represents ALL combinations of cues in Table 3 (Geometric x shading x silhouette x occlusion x ... ). But I understand this may require too much work and I would be satisfied with a convincing textual argument.

6. (Methodology) The dilation effect used for the silhouette cue does not seem to be the correct approach as it confounds the internal texture, while keeping the silhouette mostly intact.

Action: A justification for using this cue would be helpful. Personally, I feel a better way to perturb the silhouette would be to use the mask from one object to mask out a different object. In this case, the texture, shading, edges all remain intact and only the boundary is modified.

---------

I will upgrade my rating to Accept if the authors provide convincing responses to all the above questions.
I will upgrade my rating to Borderline Accept if the authors provide convincing responses to questions 1, 4 and 5.

**Ethical Concerns:**

["NO or VERY MINOR ethics concerns only"]

**Final Justification:**

The authors adequately addressed all my questions with additional experiments and detailed explanations. I had stated in my original review that I would increase my score if the points I raised were addressed, and they were.

**Limitations:**

Yes.

**Paper Formatting Concerns:**

None.

**Quality:**

2

**Strengths And Weaknesses:**

Strengths:
1. The goal of the paper to better interpret how state-of-the-art image-to-3D methods reason about their inputs is very well motivated, and has the potential to strongly influence future research in this field.
2. The tripartite classification of recent work into regression-based, multi-view and native 3D is very helpful in placing the large number of recent papers into proper context.
3. The paper is clearly written and easy to follow.

Weaknesses:
1. The evaluation section of the paper seems weak (See below for details). Since the novel contribution of the paper is derived from its evaluation of different perturbations, I feel this is a major shortcoming.
2. The methodological design does not account for confounding factors in evaluating each independent variable (perturbation category).

---

> ### Author Rebuttal · Authors · 2025-07-31
>
> Thank you for your comprehensive and detailed review! We appreciate your suggestions greatly.
>
> Primary questions (1,4, and 5):
>
> Q1: Plans for adding different visualizations to better illustrate the data in Table 3 and 5.
>
> A1:
> We appreciate your point that Tables 3 and 5 (cue-ablation results) can be difficult to quickly interpret. In our current version, the tables present raw results to faithfully represent all our experiments. However, we completely agree with your suggestion and will include the following visualizations inspired by VBench in the final version:
> 1. Radar plots for each family of methods and individual methods. These radar plots illustrate sensitivity to different cues and contextualizes performance across regression, multi-view, and native 3D models.
> 2. A color-coded Cue x Model heatmap that summarizes per‑cue degradation across methods and datasets.
> 3. For cues with variable strength (e.g., mask dilation, occlusion), bar plots showing performance drops at different perturbation strengths.
> We believe these visualizations will clearly highlight key insights and would be very happy to hear any additional suggestions from you to further improve the clarity of our presentation.
>
> Q4: Importance of shape meaningfulness vs. generic OOD failure; Shape CutMix experiments.
>
> A4: We first clarify our claims on shape meaningfulness, and then present the results of the shape CutMix experiments suggested by the reviewer, which support our conclusion derived from Zeroverse that shape meaningfulness is indeed crucial.
>
> The reviewer suggests that the failure of the models on shape meaningfulness perturbation (Zeroverse) could be a generic OOD failure. However, our goal is precisely to identify along which dimension models fail to generalize OOD. Our specific claim in the paper is that meaningful shape is critical for single-image-to-3D generalization, whereas texture meaningfulness is not. We contrast the OOD effects of shape versus texture explicitly, showing that changing to meaningless textures(while keeping other cues unchanged) had minimal impact.
>
> Following your recommendation, we conducted several shape CutMix experiments of varying difficulty:
> 1. Half-and-half: We mix half of mesh M with the other half of mesh N to construct a new mesh from GSO meshes. This limits the distribution shift and preserves many local and global shape cues (e.g., surface smoothness, local symmetry), and also preserves a significant amount of shape meaningfulness to human perception. We show 3 variants: front-back, left-right, and top-bottom.
> 2. Standard cube-based CutMix. We follow the CutMix paper and randomly sample an axis-aligned 3D cube within the bounding cube of the object. We replace the part of mesh M that falls into the cube with the part from mesh N that falls into the same cube. When sampling the cube, we pin one of its corners at the corner of the object bounding cube to avoid significant discontinuity in the output shape. The length ratio(length_{sampled cube}/length_{bounding cube}) is uniformly sampled from [0.4, 0.6]. Most parts of the object M are outside the chosen cube and remain intact. Meanwhile, the local shape cues are mostly preserved.
>
> 3. OctantMix. We center each mesh and split it into 8 octants by the coordinate planes (xy, yz, and xz planes). Then we replace the part in each octant by the corresponding part from other random meshes from the same dataset. This variant still preserves the local shape cues, and it has a significantly smaller distribution gap than zeroverse compared to our original evaluation data (GSO).
>
> We show the result on the GSO dataset, compared to Zeroverse numbers for reference. We use the CDx1000 metric following Table 3, lower is better.
>
> ||OpenLRM|CRM|SF3D|InstantMesh|Hunyuan3D-2|Trellis|
> |:---|:---|:---|:---|:---|:---|:---|
> |Original|80.89|68.07|61.58|54.54|41.82|39.64|
> |Top-Bottom|93.08(+12.19)|84.05(+15.98)|78.05(+16.47)|80.16(+25.62)|57.94(+16.12)|67.32(+27.68)|
> |Front-Back|88.96(+8.07)|80.79(+12.72)|70.89(+9.31)|69.26(+14.72)|51.74(+9.92)|58.95(+19.31)|
> |Left-Right|90.97(+10.08)|84.44(+16.37)|75.17(+13.59)|80.92(+26.38)|60.56(+18.74)|72.23(+32.59)|
> |CutMix|89.33(+8.44)|78.31(+10.24)|69.77(+8.19)|73.93(+19.39)|59.43(+17.61)|65.36(+25.72)|
> |OctantMix|97.35(+16.46)|89.96(+21.89)|92.43(+30.85)|89.49(+34.95)|75.05(+33.23)|79.94(+40.30)|
> |Zeroverse|96.59(+15.70)|81.45(+13.38)|90.34(+28.76)|89.47(+34.93)|78.09(+36.27)|78.14(+38.50)|
>
> The results show that all variants significantly harm performance. Notably, OctantMix causes a performance drop similar to zeroverse despite a smaller domain gap. Standard CutMix, which modifies only about 1/8 of the mesh volume, still results in large drops (e.g., 20 points for Hunyuan3D-2). Even minimal half-and-half perturbations, where shape meaningfulness is mostly preserved, typically cause performance drops of over 10 points, which is greater than for most other cues. This shows that shape meaningfulness is crucial, since models struggle to generalize even with partial perturbations of this cue, especially compared to other cues like texture meaningfulness.
>
> Q5: How to ensure that each individual cue is entirely isolated? Analysis on the correlation of cues.
>
> A5: Great question!
> We would like to clarify the principles behind our cue selection and provide additional quantitative analysis on correlations between the effects of different cues.
>
> We chose these cues primarily based on their perceptual importance and interpretability to humans, rather than strict orthogonality. As noted in Section 2 (Lines 104-111), our selected cues originate from psychological studies of human visual perception and have strong foundations in prior vision research, as they represent factors that humans typically find meaningful. While some cues naturally remain disentangled (such as shading versus texture), others inherently overlap to some extent (like style and texture, as you correctly pointed out).
>
> We appreciate your suggestion to quantitatively analyze cue correlations. However, some perturbations, such as style and texture, are difficult to combine meaningfully, making full combinational analyses like PCA impractical. Instead, we assess whether cues impact objects similarly by calculating per-object performance drops in CD for each cue and then computing the Spearman rank correlation between pairs of cues. This produces a correlation matrix showing how similarly each pair of cues affects the same set of objects. Note that this analysis only correlates with cue correlation but does not directly indicate cue correlation.
>
> ||Texture|Shading|Silhouette|Occlusion|Edges|Local continuity|Style|
> |:---|:---|:---|:---|:---|:---|:---|:---|
> |**Texture**|**1.00**|0.66|0.31|0.29|0.36|0.39|0.50|
> |**Shading**|0.66|**1.00**|0.34|0.35|0.35|0.51|0.60|
> |**Silhouette**|0.31|0.34|**1.00**|0.27|0.24|0.28|0.34|
> |**Occlusion**|0.29|0.35|0.27|**1.00**|0.19|0.30|0.31|
> |**Edges**|0.36|0.35|0.24|0.19|**1.00**|0.27|0.31|
> |**Local continuity**|0.39|0.51|0.28|0.30|0.27|**1.00**|0.47|
> |**Style**|0.50|0.60|0.34|0.31|0.31|0.47|**1.00**|
>
>
> The result suggests that overall the correlation is low. Interestingly, texture and shading cues seem to affect a set of objects in similar ways, though they are inherently disentangled.
>
> Other questions (2, 3, 6):
>
> Q2: Provide some statistical tests to show that the drop in metrics are actually significant.
>
> A2: Following your valuable suggestion, in the final version, we will include statistical tests (specifically, Wilcoxon signed-rank tests) to clearly demonstrate that the observed drops in performance metrics are statistically significant.
>
> Q3: How is the performance robustness of each cue determined in Fig.1 (right)?
>
> A3: The radar plot in Figure 1 illustrates normalized increases in Chamfer Distance (CD) when using perturbed images relative to original images. Specifically, each axis is normalized from 0 to 1 based on the largest observed drop across all models and cues. Detailed descriptions for how we compute each axis are provided in Appendix Section 2.3.
>
> Q6: The dilation effect for the silhouette cue confounds internal texture; instead, use the mask from one object to mask out a different object.
>
> A6: Thank you for highlighting this! We agree this is a valid concern, as dilated regions were filled with solid black. We will explain our design decision and provide results following your recommended approach.
> In our dilation perturbation, no image cue is removed, only black space is added. If the model ignored silhouette cues, it would rely on other cues and reproduce the original shape, which did not occur, suggesting the model depends on silhouette information.
>
> Following your suggestion, for each mesh M, we randomly sample another mesh N and combine the image content of M with the silhouette of N (scaling M to avoid blank space). We then evaluate each model’s output against both the ground truth by image content (M) and by silhouette (N). Results are provided below.
>
> ||OpenLRM|CRM|SF3D|InstantMesh|Hunyuan3D-2|Trellis|
> |:---|:---|:---|:---|:---|:---|:---|
> |Original|80.89|68.07|61.58|54.54|41.82|39.64|
> |ContentGT(M)|196.88(+115.99)|154.52(+86.45)|143.08(+81.50)|194.25(+139.71)|167.95(+126.13)|119.27(+79.63)|
> |SilhouetteGT(N)|96.31(+15.42)|98.42(+30.35)|103.55(+41.97)|85.74(+31.20)|100.96(+59.14)|102.00(+62.36)|
>
>
> Results indicate that performance significantly decreases with both ground truths. However, performance against the silhouette ground truth (N) is substantially better, clearly indicating that all models indeed rely heavily on silhouette cues to infer shape. These findings align well with our original conclusions, and we will include these results in the final version of our paper.
>
> Thank you again for your thoughtful and constructive suggestions! We also welcome your further suggestions or questions!

---

> > ### Comment · Reviewer_wyxx · 2025-08-05
> > **Raising Rating to Accept**
> >
> > I appreciate the effort put in by the authors in answering all my questions. As I mentioned in my original review I would be willing to increase my rating if my main concerns were adequately addressed. This has been done, and I'm upgrading my rating to ACCEPT.

---

### Decision · Program_Chairs · 2025-09-17

**Decision:**

Accept (spotlight)

**Comment:**

This paper introduces Cue3D, a comprehensive and model-agnostic framework for quantifying how modern single-image 3D generation models use classic visual cues like shading, texture, and silhouette. The work is very well-motivated, and the proposed benchmark and analysis provide valuable insights into the inner workings of state-of-the-art models, revealing key dependencies on geometric cues and shape meaningfulness. All reviewers converged on an "Accept" recommendation after an exceptionally thorough and constructive discussion period, where the authors provided extensive new experiments and clarifications that successfully addressed all initial concerns.